# BIGCHARTS-R1: Enhanced Chart Reasoning with Visual Reinforcement Finetuning

**Ahmed Masry**[1,2,*]**, Abhay Puri**[1]**, Masoud Hashemi**[1]**, Juan A. Rodriguez**[1,3,5]**,
Megh Thakkar**[1]**, Khyati Mahajan**[1]**, Vikas Yadav**[1]**, Sathwik Tejaswi Madhusudhan**[1]**,
Alexandre Piché**[1]**, Dzmitry Bahdanau**[1,3,7]**, Christopher Pal**[1,3,6,7]**, David Vazquez**[1]
**Enamul Hoque**[2]**, Perouz Taslakian**[1]**, Sai Rajeswar**[1,3,4]**, Spandana Gella**[1]

[1]ServiceNow Research  [2]York University  [3]Mila  [4]Université de Montréal
[5]ÉTS Montréal  [6]Polytechnique Montréal  [7]CIFAR AI Chair

## Abstract

Charts are essential to data analysis, transforming raw data into clear visual representations that support human decision-making. Although current vision-language models (VLMs) have made significant progress, they continue to struggle with chart comprehension due to training on datasets that lack diversity and real-world authenticity, or on automatically extracted underlying data tables of charts, which can contain numerous estimation errors. Furthermore, existing models only rely on supervised fine-tuning using these low-quality datasets, severely limiting their effectiveness. To address these issues, we first propose BIGCHARTS, a dataset creation pipeline that generates visually diverse chart images by conditioning the rendering process on real-world charts sourced from multiple online platforms. Unlike purely synthetic datasets, BIGCHARTS incorporates real-world data, ensuring authenticity and visual diversity, while still retaining accurate underlying data due to our proposed replotting process. Additionally, we introduce a comprehensive training framework that integrates supervised fine-tuning with Group Relative Policy Optimization (GRPO)-based reinforcement learning. By introducing novel reward signals specifically designed for chart reasoning, our approach enhances model robustness and generalization across diverse chart styles and domains, resulting in a state-of-the-art chart reasoning model, BIGCHARTS-R1. Extensive experiments demonstrate that our models surpass existing methods on multiple chart question-answering benchmarks compared to even larger open-source and closed-source models.

## 1 Introduction

Charts are essential tools for transforming raw data into clear and visually intuitive formats, significantly aiding data analysis, interpretation, and decision-making across numerous domains such as scientific publications, business reporting, and media presentations (Kim et al., 2020). Given their ubiquity, there have been increasing efforts into developing models capable of understanding and reasoning about charts (Hoque et al., 2022).

Existing methods aimed at training these vision-language models (VLMs) capable of chart comprehension predominantly use supervised finetuning (SFT) by utilizing synthetic datasets generated by teacher models, primarily due to the high cost of manual annotation (Liu et al., 2024b; Meng et al., 2024; Zhang et al., 2024; Masry et al., 2023). These methods typically follow a two-step process: (i) generate synthetic chart images, and (ii) generate corresponding question-answer (QA) pairs from language models such as Gemini (Georgiev et al., 2024) or GPT4 (OpenAI et al., 2024). Despite their practicality, such

---

*Correpondance to ahmed.masry@servicenow.com and spandana.gella@servicenow.com

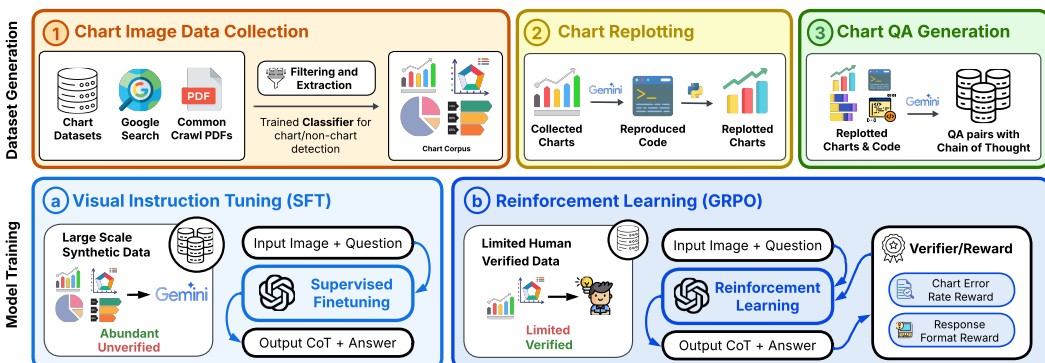

Figure 1: **BIGCHARTS construction and BIGCHARTS-R1 training pipeline.** We begin by extracting a high-quality corpus from open chart datasets, Google Search, and Common Crawl(§3.1.1). We then generate and execute the code responsible for producing these charts to replot them (§3.1.2), and then derive question-answer pairs with chain-of-thought reasoning (§3.1.3). For training BIGCHARTS-R1, we use a two-stage approach: (i) visual instruction tuning via SFT on large-scale synthetic data(§4.1), and (b) RL (GRPO) with verifiable rewards and human-labeled data to enhance chart reasoning (§4.2).

methods have significant drawbacks. First, synthetic charts created *purely* with plotting tools (e.g., matplotlib) using a narrow set of controlled parameters are visually homogeneous and fail to replicate the rich diversity observed in real-world charts. Second, synthetic QA pairs generated from plotting scripts often accurately reflect underlying data but neglect crucial visual details of the chart images (see Figure 2b) (Liu et al., 2024b; Han et al., 2023; Meng et al., 2024; Deitke et al., 2024; Yang et al., 2025). This can lead to models overfitting to certain types of charts and chart components, affecting their generalization and effectiveness when encountering visually different charts (Wang et al., 2024).

An alternative line of research has sought to leverage chart images directly crawled from real-world sources to capture visual variety (Masry et al., 2024a;b). However, these images typically lack access to accurate underlying data (e.g., plotting code), necessitating reliance on external models to estimate data values directly from the chart images. This leads to substantial inaccuracies and estimation errors (Wang et al., 2024) (see Figure 2a) in the dataset, subsequently affecting the model training and downstream performance.

To summarize, these limitations-—visual homogeneity of the dataset, estimation errors in underlying chart data, and reliance on simple SFT prone to overfitting-—significantly hinder model performance on in-domain tasks and impair generalization to out-of-distribution scenarios for chart understanding and reasoning (Kumar et al., 2025; Trung et al., 2024). To overcome these limitations, we present two novel contributions: a pipeline to curate a diverse and accurate dataset for training chart reasoning models, and a training strategy utilizing the curated dataset. Firstly, we introduce BIGCHARTS, a novel dataset and pipeline explicitly designed for enhanced chart reasoning as shown in Figure 1. Our innovative method combines the strengths of real-world and synthetic datasets by collecting visually diverse real-world charts and conditionally replotting them using VLMs. While initial VLM-generated code for the charts may have inaccuracies, we mitigate this by re-rendering these codes into new, improved chart images and discarding the original charts, ensuring visual authenticity coupled with accurate underlying data. Furthermore, in the subsequent QA generation stage, we provide both the rendered charts and their associated data in the form of code, enabling VLMs to create logically consistent QA pairs accurately reflecting both visual and numerical characteristics (Figure 2c).

We also propose a robust training strategy that integrates SFT with reinforcement learning (RL), leveraging Group Relative Policy Optimization (GRPO) (Shao et al., 2024) as it has proven effective in improving reasoning in textual contexts. We introduce novel reward functions specifically designed to encourage models to leverage visual chart properties for accurate numerical estimation and mathematical and visual reasoning. Our extensive experiments demonstrate that the models trained with BIGCHARTS and method,

BigCharts-R1, significantly surpass existing state-of-the-art approaches across multiple chart reasoning benchmarks and exhibit better generalization to out-of-distribution tasks.

Our main contributions are: (1) BigCharts: A visually diverse, high-quality chart training dataset possessing precise ground-truth data alongside realistic visual variety; (2) BigCharts-R1: Advanced chart reasoning models that combine SFT and GRPO using novel chart-specific rewards, achieving state-of-the-art performance and robust generalization in both 3B and 7B model categories; and (3) Extensive empirical evaluations confirming the effectiveness and robust generalization advantages of our dataset and training methodology. We release our code and artifacts at bigcharts.github.io.

## 2  Related Work

**Instruction-tuning for Chart Understanding.**  There have been development of models exclusively for chart understanding (Masry et al., 2023; Liu et al., 2023; Zhou et al., 2023; Masry et al., 2025), as well as models that are obtained by finetuning existing VLMs on chart understanding datasets (Masry et al., 2024a; Han et al., 2023; Meng et al., 2024; Masry et al., 2024b; Rodriguez et al., 2025). In the second set of works, datasets used to fine-tune VLMs such as ChartLlama (Han et al., 2023), ChartAssistant (Meng et al., 2024), and ChartInstruct (Masry et al., 2024a) rely on underlying datatables of the charts either obtained from the original dataset source or extracted from the charts, failing to capture the nuances of the chart images. Methods like ChartGemma (Masry et al., 2024b) utilize the chart images directly and use closed-source models to generate the datatable for curating instruction-tuning samples. This leads to data estimation errors introduced by the multi-modal models, impairing the quality of the training dataset. Our dataset addresses these critical concerns as it consists of a large variety of charts that have faithful underlying datatables, enabling effective and accurate learning.

**Post-training for Reasoning in LLMs and its extension to VLMs.**  Numerous recent post-training methods for LLMs have focused on enhancing their multi-step reasoning abilities (Kumar et al., 2025). This is achieved by leveraging reasoning traces in the dataset and sophisticated extensions of RL or SFT methods. Initial RL-based approaches, such as REINFORCE (Williams, 1992), treated token generation as sequential decision-making but struggled with high variance. Newer methods like Proximal Policy Optimization (PPO; Schulman et al., 2017a) introduced clipped policy updates, significantly improving training stability for extended reasoning tasks. Further refinements include Direct Preference Optimization (DPO; Rafailov et al., 2024) and Group Relative Policy Optimization (GRPO; Shao et al., 2024), which directly optimize outputs to human preferences through binary comparisons or relative rankings, respectively, thereby enhancing reasoning coherence. In parallel, chain-of-thought finetuning (Chung et al., 2022) explicitly trains models to generate intermediate reasoning steps, improving interpretability and logical consistency. Complementing these approaches, recent studies have also leveraged synthetic reasoning datasets to augment training sets, further improving generalization on reasoning-intensive tasks (Kim et al., 2023). These developments in LLMs have recently been adapted for finetuning VLMs (Liu et al., 2025a), but their effectiveness in tasks such as chart comprehension, which requires strong mathematical and visual reasoning abilities, is underexplored.

**Chart Benchmarks.**  A variety of tasks and benchmarks have been designed to assess VLMs' chart understanding capabilities, including question answering (Masry et al., 2022; Wang et al., 2024), chart summarization (Kantharaj et al., 2022b), fact-checking (Akhtar et al., 2023a;b), explanation and caption generation (Kantharaj et al., 2022a; Tang et al., 2023), and visualization recommendation systems (Hu et al., 2019). Early benchmarks like FigureQA (Kahou et al., 2018), DVQA (Kafle et al., 2018), Leaf-QA (Chaudhry et al., 2019), and PlotQA (Methani et al., 2020) were based on synthetically generated charts and templated questions, and had limited visual diversity. More recent benchmarks, such as ChartQA (Masry et al., 2022) and CharXiv (Wang et al., 2024) incorporate real-world charts and introduce more complex questions, requiring advanced visual reasoning.

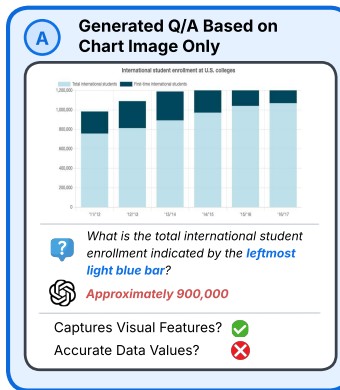 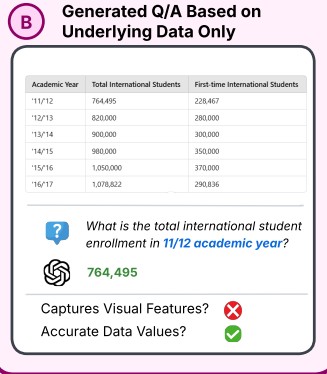 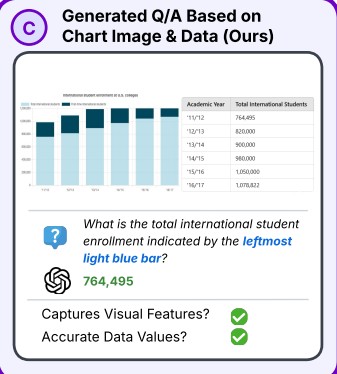

Figure 2: **Different Chart Generation Approaches**. Generating QA pairs automatically can be achieved with three approaches: using only images (*image-only*), leveraging raw data values from the charts (*data-only*), or combining both (*our proposed approach*). Our method integrates semantic and visual features while ensuring accuracy in underlying data values.

## 3 BIGCHARTS Dataset

### 3.1 Generation Pipeline

As illustrated in Figure 1, our BIGCHARTS pipeline for curating the dataset comprises of three stages: *(1) Chart Collection*, *(2) Chart Re-Plotting*, and *(3) Question and CoT Generation*.

#### 3.1.1 Chart Collection

We initially collect 245,414 chart images from three diverse sources to ensure comprehensive visual and topical coverage:

1. **Existing Datasets.** We aggregate images from existing datasets - ChartGemma (Masry et al., 2024b), FigureQA (Kahou et al., 2018), DVQA (Kafle et al., 2018), PlotQA (Methani et al., 2020), and ArXivQA (Li et al., 2024b), resulting in a total of 174,277 images. While these datasets form a solid base, visual diversity varies significantly among them.

2. **Common Crawl.** To enhance visual diversity, we utilize the Mint-1T dataset (Awadalla et al., 2024), which includes extensive PDF files covering numerous topics from Common Crawl[1]. We extract images from these documents and employ a rigorous two-step filtering approach to obtain charts. We first train a high-recall ResNet-50 (He et al., 2015) binary classifier using labeled chart images (positive class) and natural images from CC-12M (Changpinyo et al., 2021) and ImageNet (Russakovsky et al., 2015) (negative class). Then, we use this classifier to select potential chart images. In the second stage, we enhance the classifier by manually labeling 5,000 predicted charts from the first stage and retrain the classifier aiming for higher precision. Overall, this process yields 57,196 filtered charts. Despite broad topical coverage, prevalent types remain bar, line, and pie charts. Detailed methodology used for filtering Mint-1T is available in Appendix A.2.

3. **Google Search.** To further diversify chart styles, such as heatmaps, scatter plots, dashboards, and infographics, we conduct targeted Google searches using 200 curated keywords. We collect the top search results for each keyword, acquiring an additional 13,941 visually diverse charts. The full keyword list is provided in Appendix A.3.

#### 3.1.2 Chart Re-Plotting

A critical challenge in chart reasoning datasets derived from real-world images is the absence of underlying data (e.g., plotting code or datatables). To address this, we introduce a

---

[1]https://commoncrawl.org/

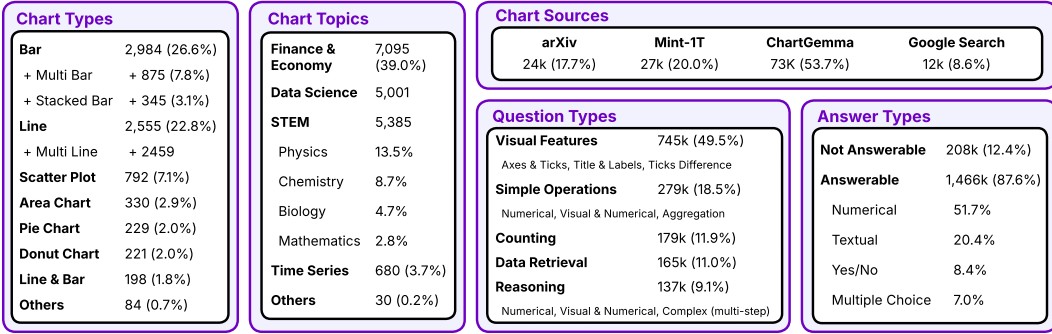

Figure 3: **BIGCHARTS Dataset Statistics.** Type and topic distribution of the charts, chart sources, and type distribution of questions and answers.

novel *chart re-plotting* strategy that recovers both visual styles and underlying data. We utilize Gemini Flash 2.0 (Georgiev et al., 2024) to generate code replicating the design and content of each chart image. Recognizing potential inaccuracies in data estimation by vision-language models (VLMs) (Wang et al., 2024), we subsequently render new chart images directly from this generated code, ensuring visual authenticity and data consistency.

We predominantly employ Python-based plotting libraries, `matplotlib` and `plotly`, for most collected charts. However, for more stylistically complex charts from ChartGemma and Google Search, we leverage the Chart.js[2] library which captures their intricate designs more accurately. Charts resulting in rendering failures due to erroneous code are excluded, ultimately resulting in a refined set of 134,950 charts complete with accurate underlying data. Detailed prompts for code generation are described in Appendix A.4. We provide samples of original and replotted charts in Figure 19 in A.8.

### 3.1.3 Question and Chain-of-Thought (CoT) Generation

After curating our chart images, we generate corresponding questions enriched with detailed step-by-step reasoning. Unlike prior methods reliant solely on either underlying data (missing visual context) or visual chart images alone (prone to data inaccuracies), our approach integrates both chart images and their accurate underlying data for question generation (see Figure 2). This combined approach enables VLMs to produce QA pairs that accurately reflect both visual and numerical chart characteristics, significantly reducing errors and improving overall dataset quality.

For each chart, we use Gemini Flash 2.0) to generate sixteen questions – targeting direct data retrieval, visual analysis, and mathematical reasoning – accompanied by a comprehensive chain-of-thought reasoning process leading to final answers. Detailed prompt instructions and examples are provided in A.4 and A.6, respectively.

### 3.2 Dataset Analysis

Figure 3 provides statistics of the BIGCHARTS dataset, presenting the distribution of chart sources, chart topics, and chart question and answer types. We provide the prompts used to obtain our statistics in Appendix A.5.

**Chart Types and Topics** We use Gemini Flash 2.0 to classify the charts into various types and observe that the distribution demonstrates a variety of visual representations. While bar charts, line charts, and multi-line charts are the most common, the dataset also includes specialized visualizations such as scatter plots, stacked bar charts, and area charts. The "Other" category includes less frequent but important visual types like heatmaps, radar charts, waterfall charts, and violin plots.

---

[2]https://www.chartjs.org/

For the topic distribution, we initially prompt Gemini Flash 2.0 using the chart images to obtain topics, and then use GPT-4o to cluster them into 20 representative groups based on their similarity. *Finance & economy (39%)*, *STEM (29.7%)*, and *Data Science (27.5%)* constitute the largest portion of the dataset, encompassing a broad range of subtopics. Finance includes subtopics like GDP, stock market trends, and inflation. Data Science covers statistical methods, machine learning techniques, and time series analysis, which are widely used across multiple domains for data-driven decision-making. STEM topics like *Physics* and *Chemistry* reflect the dataset's applicability to scientific research.

**Questions and Answers in Dataset Samples**  Each dataset sample contains three elements: 1) question, 2) answer, and 3) chain-of-thought. As shown in Figure 3, BIGCHARTS consists of a total of around 1.8 million questions across various question types. The majority of questions are about *visual features*, asking the title, information in specific positions, axes, labels, etc. Other question types include retrieving numerical data from the images, doing mathematical computations, and performing single-step or multi-step reasoning. BIGCHARTS also has unanswerable questions with "Not Applicable" as the ground truth to test a model's ability to abstain and recognize unanswerable questions. Answers also have different types: numerical, textual, multiple-choice options, and abstention for unanswerable questions. Among the answerable questions, *numeric answers* are the most prevalent, accounting for over half of the dataset. This is followed by 342k *textual answers* and 141k *Yes/No* questions. *Multiple-choice* questions with options make up almost 118k samples.

Each question has a generated chain-of-thought (CoT), which is used for model finetuning. These CoTs vary in length, with the longest containing 1,933 tokens and a median length of 39 tokens (and an average of 49.65 tokens), indicating that the majority of CoTs are relatively concise. We present the distribution of the length of the CoT in Figure 5.

## 4  Methodology

We employ the Qwen2.5-VL-Instruct (Bai et al., 2025) models with 3B and 7B parameters for our experiments, chosen due to their state-of-the-art capabilities in vision-language understanding tasks. Our training methodology comprises two primary stages: supervised finetuning (SFT), followed by reinforcement learning (RL) using Group Relative Policy Optimization (GRPO). We present an overview of our training pipeline in Figure 1.

### 4.1  Supervised Finetuning (SFT)

We perform supervised finetuning using the high-quality reasoning chain-of-thought data from our BIGCHARTS dataset. This improves the model's ability to extract precise data values from chart images while encouraging detailed, step-by-step reasoning, enhancing the model's overall capability in solving complex chart-based problems. We provide the prompts used for finetuning our models in Appendix A.4.

### 4.2  Reinforcement Learning (RL)

To mitigate potential errors arising during the SFT phase where inaccuracies from teacher models can propagate and degrade model performance, we employ RL. This step is designed to rectify deviations resulting from imperfect training data, consequently enhancing the model's chart understanding and reasoning capabilities. Our RL approach follows the Reinforcement Learning with Verifiable Rewards (RLVR) framework (Lambert et al., 2025), specifically utilizing Group Relative Policy Optimization (GRPO) (Shao et al., 2024) given its recently proven effectiveness in enhancing reasoning in LLMs.

**Reinforcement Learning with Verifiable Rewards (RLVR)**  RLVR has recently been applied for training large language models through reinforcement learning on tasks with verifiable outcomes, such as mathematics and code generation (Zelikman et al., 2022; Liu et al., 2025b). Unlike traditional methods like Reinforcement Learning from Human Feedback (RLHF) (Ouyang et al., 2022; Christiano et al., 2023), which rely on learned reward

models, RLVR directly employs verifiable reward functions ($R$) that objectively assess outcomes. The RLVR training objective is defined as follows:

$$\max_{\pi_\theta} \mathbb{E}_{y \sim \pi_\theta(x)} \left[ R(x, y) \right] = \mathbb{E}_{y \sim \pi_\theta(x)} \left[ R(x, y) - \beta KL \left[ \pi_\theta(y|x) \| \pi_{\text{ref}}(y|x) \right] \right], \quad (1)$$

where $R$ is the reward, $\beta$ is a scaling factor, and $KL$ represents the $KL$ divergence between the current model policy $\pi_\theta$ and the reference model policy $\pi_{\text{ref}}$.

**Group Relative Policy Optimization (GRPO)**   A primary advantage of GRPO is its independence on a separate value or critic model, unlike PPO (Schulman et al., 2017b). By doing so, GRPO substantially reduces memory usage and improves training efficiency by estimating values directly from sampled response groups.

Given an input query $q$, GRPO samples a set of responses $o_1, o_2, \ldots, o_G$ from the current policy model $\pi_{\sigma_{\text{old}}}$. Each response is evaluated by our reward functions to yield individual rewards $r_1, r_2, \ldots, r_G$. The advantage $A_i$ for each response is then computed as:

$$A_i = \frac{r_i - mean(r_1, r_2, ..., r_G)}{std(r_1, r_2, ..., r_G)} \quad (2)$$

**Reward Function Formulation**   We propose two complementary reward signals: *Chart Error Rate Reward (CERM)* and *Response Format Reward*, which are combined to guide the reinforcement learning process.

The *Chart Error Rate Reward (CERM)* is a smooth, error-based dense reward specifically designed for numeric answers. We first calculate an *error rate* (ER) based on the difference between the predicted value and the ground-truth value . Then, we apply a homographic function to map this error rate into a continuous reward between $(0, 1]$, incentivizing the model to minimize numerical prediction errors. For non-numeric answers, we utilize a straightforward exact-match criterion:

$$\text{ER}(\hat{y}, y) = \frac{|\hat{y} - y|}{|y|}, \qquad R_{\text{CERM}}(\hat{y}, y) = \begin{cases} \dfrac{1}{1 + \text{ER}(\hat{y}, y)}, & \text{if both } \hat{y} \text{ and } y \text{ are numeric,} \\ 1, & \text{if non-numeric and } \hat{y} = y, \\ 0, & \text{otherwise.} \end{cases} \quad (3)$$

Additionally, inspired by Shen et al. (2025), we define a *Response Format Reward* ($R_{\text{Fmt}}$) to encourage adherence to the required answer format:

$$R_{\text{Fmt}} = \begin{cases} 1 & \text{if the model follows the required response structure,} \\ 0 & \text{otherwise.} \end{cases} \quad (4)$$

The final combined scalar reward is computed as the sum of these two signals:

$$R_{\text{total}} = R_{\text{CERM}} + R_{\text{Fmt}} \quad (5)$$

## 5   Experiments and Results

### 5.1   Experimental Setup

**Training Setup.**   For our SFT experiments, we use LlamaFactory (Zheng et al., 2024). Both the 3B and 7B models are trained for one epoch on BIGCHARTS with SFT using a learning rate of 2e-5, batch size of 32, 0.1 warmup ratio, and a cosine scheduler.

For GRPO training, we train for one epoch with a batch size of 8, learning rate of 1e-6, and generate 8 candidate outputs per sample. All experiments are conducted on a single node with 8×H100-80GB GPUs. To construct our RL data, we utilize a combination of high-quality human-labeled and verified datasets such as the ChartQA training split and 1K randomly sampled instances from the training subsets of the template-based datasets: PlotQA (v1 & v2), DVQA, and FigureQA. Overall, our RL dataset contains 22,297 chart images and 32,297 questions with verifiable answers.

| Model | FigureQA-Sub | | DVQA-Sub | | PlotQA-Sub | | ChartQA | | | CharXiv | | Avg |
|---|---|---|---|---|---|---|---|---|---|---|---|---|
| | Val1 | Val2 | ValE | ValH | T1 | T2 | aug | hum | avg | Reas. | Des. | |
| **Closed-Source Models** | | | | | | | | | | | | |
| GPT-4o (OpenAI et al., 2024) | **65.70** | **69.10** | 57.50 | **61.20** | 59.50 | 19.90 | - | - | 85.07 | 50.50 | 82.58 | **61.22** |
| Gemini-Flash-2.0 (Georgiev et al., 2024) | 54.90 | 54.50 | **60.60** | 59.50 | **60.40** | 32.70 | - | - | 85.40 | 50.30 | 75.10 | 59.26 |
| Claude Sonnet 3.5 (Anthropic, 2024) | 43.30 | 44.70 | 56.90 | 56.60 | 49.20 | **32.90** | - | - | **90.80** | **60.20** | **84.30** | 57.65 |
| **Open-Source Models < 7B** | | | | | | | | | | | | |
| Intern-VL2.5-1B (Chen et al., 2025) | 59.4 | 60.0 | 93.2 | 92.2 | 61.70 | 24.80 | - | - | 75.9 | 19.00 | 38.40 | 58.29 |
| Intern-VL2.5-2B (Chen et al., 2025) | 64.3 | 64.3 | **97.5** | **95.7** | **71.10** | **38.20** | - | - | 79.2 | 21.30 | 49.70 | **64.59** |
| Phi 3.5-Vision-4B (Abdin et al., 2024) | **64.9** | **66.8** | 84.9 | 84.1 | 48.6 | 11.90 | - | - | **81.8** | **32.70** | **55.02** | 58.97 |
| **Open-Source Models 7-12B** | | | | | | | | | | | | |
| Intern-VL2.5-8B (Chen et al., 2025) | **69.60** | **69.00** | **96.60** | **95.20** | **74.70** | **42.30** | - | - | **84.80** | **32.90** | **68.60** | **70.41** |
| LLaVA-Next-Mistral-7B Li et al. (2024a) | 58.1 | 57.7 | 72.1 | 71.2 | 41.7 | 8.0 | - | - | 51.80 | 13.90 | 35.40 | 45.54 |
| Llama 3.2-Vision-11B (Grattafiori et al., 2024) | 0.0 | 0.0 | 3.5 | 3.2 | 0.0 | 0.0 | - | - | 83.40 | 31.20 | 59.35 | 20.07 |
| **Chart-Specific LVLMs** | | | | | | | | | | | | |
| ChartGemma-3B (Masry et al., 2024b) | 38.90 | 37.00 | 37.90 | 37.00 | 35.60 | 20.70 | 90.80 | 69.52 | 80.16 | **12.50** | **21.30** | 35.67 |
| TinyChart-3B (Zhang et al., 2024) | **48.00** | **46.10** | **61.90** | **50.20** | **55.30** | **50.60** | **93.86** | **73.34** | **83.60** | 8.30 | 16.15 | **46.68** |
| **Our Qwen2.5-VL Models** | | | | | | | | | | | | |
| Qwen2.5-VL-3B (CoT) | 58.10 | 57.00 | 76.20 | 75.60 | 54.80 | 43.30 | 86.40 | 63.84 | 75.12 | 32.60 | 59.77 | 59.17 |
| Qwen2.5-VL-3B + SFT | 76.10 | 75.70 | 76.30 | 73.80 | 74.60 | 58.40 | 90.00 | 79.20 | 84.60 | 36.00 | **62.85** | 68.71 |
| BigCharts-R1-3B | **80.10** | **81.00** | **81.20** | **80.60** | **78.50** | **59.90** | **94.32** | **82.00** | **88.16** | **37.40** | 62.38 | **72.14** |
| Qwen2.5-VL-7B (CoT) | 80.70 | 79.30 | 78.30 | 78.30 | 73.40 | 50.40 | 81.68 | 71.28 | 76.48 | 41.30 | 66.85 | 69.45 |
| Qwen2.5-VL-7B + SFT | 79.10 | 75.90 | 79.80 | 77.50 | 77.70 | 60.40 | 91.44 | 80.88 | 86.16 | 39.40 | **69.00** | 71.66 |
| BigCharts-R1-7B | **81.20** | **81.20** | **83.80** | **83.60** | **80.90** | **61.90** | **94.88** | **84.80** | **89.84** | 41.30 | 66.58 | **74.48** |

Table 1: Comparison of BIGCHARTS-R1 and its variants with open-source and closed-source baselines on chart question answering benchmarks (§5.2). Color coding for comparing categories:  closed-source models ,  open-source models below 7B paramaters , and open-source models between 7-12B parameters .  Best scores in each category are presented in bold.

**Evaluation Benchmarks.**   We evaluate our method and models on a diverse set of chart QA benchmarks , which encompasses both real-world and synthetic datasets.  For real-world evaluations, we use ChartQA (Masry et al., 2022), and CharXiv (Wang et al., 2024). We also include three synthetic benchmarks—FigureQA (Kahou et al., 2018), DVQA (Kafle et al., 2018), and PlotQA (Methani et al., 2020)—which also serve as effective measures of reasoning abilities.  These synthetic datasets often have extraordinarily large test and validation splits (for instance, PlotQA's Test1 set contains 1.1M Q/A pairs derived from just 74 templates), making full-scale evaluations overly expensive. To alleviate this, we create *FigureQA-Sub*, *DVQA-Sub*, and *PlotQA-Sub* by sampling 1K chart QA pairs from the original splits. These smaller subsets preserve the diversity of both visual layouts and question types while significantly reducing the computational burden.  We plan to publicly release these subsets, offering the research community a practical and representative suite of chart QA evaluation benchmarks to evaluate a wide range of chart understanding abilities.

**Evaluation Metrics.**   For our evaluations, we use exact accuracy as the metric for *FigureQA-Sub* and *DVQA-Sub*, while we use relaxed accuracy  (Methani et al., 2020; Masry et al., 2022) for *PlotQA-Sub* and *ChartQA*. For CharXiv, we use GPT4o in conjunction with the prompt proposed in the original work (Wang et al., 2024) for evaluation.

## 5.2   Performance Comparison on Chart Question Answering Benchmarks

**Synthetic Benchmarks:**   We compare the performance of various open-source and closed-source baselines with our models across different chart question answering benchmarks in Table 1. Performance of baseline models on synthetic benchmarks such as FigureQA-Sub, DVQA-Sub, and PlotQA-Sub is relatively low.  This is mainly because these datasets contain charts that lack explicit numerical labels on their visual elements (Figures 15, 16, 17), forcing models to interpolate numerical values visually.  In contrast, charts in the ChartQA dataset typically include clear numerical labels (Figure 14), simplifying the model's recognition task.  Previous model evaluations on ChartQA have not sufficiently addressed this issue, which we highlight here to encourage further research.

Our results show that fine-tuning the Qwen2.5-VL models with BIGCHARTS improves performance on the synthetic benchmarks, particularly when combined with the reinforce-

| Model (SFT Dataset) | FigureQA-Sub | | DVQA-Sub | | PlotQA-Sub | | ChartQA | CharXiv | | Avg |
|---|---|---|---|---|---|---|---|---|---|---|
| | Val1 | Val2 | ValE | ValH | T1 | T2 | avg | Reas. | Des. | |
| BIGCHARTS vs. Existing Datasets | | | | | | | | | | |
| Qwen2.5-VL-3B (TinyChart) | 47.30 | 48.80 | 51.40 | 48.19 | 57.09 | 50.00 | 70.28 | 24.60 | 41.62 | 48.80 |
| Qwen2.5-VL-3B (ChartGemma) | 78.30 | 79.10 | 73.30 | 74.30 | 40.50 | 58.60 | 68.48 | 10.70 | 39.25 | 58.05 |
| Qwen2.5-VL-3B (BIGCHARTS) | 76.10 | 75.70 | 76.30 | 73.80 | 74.60 | 58.40 | 84.60 | 36.00 | 62.85 | 68.70 |
| Original Charts vs. Replotted Charts & Code | | | | | | | | | | |
| Qwen2.5-VL-3B (Original Charts) | 73.70 | 73.10 | 75.20 | 72.50 | 72.40 | 52.40 | 82.24 | 30.90 | 63.62 | 66.22 |
| Qwen2.5-VL-3B (Replotted Charts) | 76.10 | 75.70 | 76.30 | 73.80 | 74.60 | 58.40 | 84.60 | 36.00 | 62.85 | 68.70 |

Table 2: Ablation results comparing *(i)* Qwen2.5-VL-3B fine-tuned on BIGCHARTS vs. existing datasets (TinyChart and ChartGemma), and *(ii)* training on Q/A pairs generated from original charts vs. our replotted charts with code. BIGCHARTS consistently yields stronger performance, particularly on challenging benchmarks like ChartQA, PlotQA, and CharXiv.

ment learning step. We attribute this to the fact that our novel approach of integrating chart images and corresponding data tables during the SFT data generation enables accurate numerical retrieval and visual interpolation in our models (illustrated in Figure 2).

**BigCharts-R1:** Supervised finetuning (SFT) significantly enhances the model's ability to accurately interpret chart images and retrieve precise numerical data, while also strengthening its reasoning capabilities. However, SFT alone can propagate errors introduced by the teacher models used during curating the finetuning data. Reinforcement Learning (RL), specifically through our novel reward function (Chart Error Rate Reward, CERM), mainly boosts reasoning performance rather than simple data retrieval. This distinction is evident in the descriptive (Des.) subset of the CharXiv benchmark, which emphasizes simple information retrieval without complex reasoning. Here, RL does not enhance, but infact slightly decreases the performance, unlike other benchmarks which include numerous reasoning-intensive questions benefiting from our RL approach.

Our results show that combining SFT and RL effectively balances improvements in chart comprehension, understanding, and advanced reasoning capabilities. Notably, improvements are more pronounced for the 3B model compared to the 7B model, suggesting potential saturation in larger models and highlighting the need for stronger vision capabilities in backbone models through improved pretraining strategies. Overall, BIGCHARTS-R1 sets a new state-of-the-art standard for chart reasoning across multiple established chart question-answering benchmarks.

## 5.3 Ablation Studies

**BigCharts vs Existing Chart Datasets.** To ensure a fair comparison between BIGCHARTS and prior datasets such as TinyChart (Zhang et al., 2024) and ChartGemma (Masry et al., 2024b), we fine-tuned the same backbone model (Qwen2.5-VL-3B (Bai et al., 2025)) on these datasets using identical hyperparameters (see Section 5.1). We also adopted the official codebases and "program-of-thought" formats from TinyChart and ChartGemma to ensure their optimal performance. As shown in the upper part of Table 2, fine-tuning on BIGCHARTS consistently yields superior performance, especially on challenging benchmarks like ChartQA, PlotQA, and CharXiv. These results highlight the effectiveness and broader potential of BIGCHARTS for advancing chart reasoning research.

**Replotted Charts vs. Original Charts.** We further compared models trained on Q/A pairs generated from original real-world charts with those trained on our synthetic replotted charts (which include both the chart image and underlying code). For fairness, Q/A pairs for the original charts were generated using the same teacher model, Gemini Flash 2.0 (Georgiev et al., 2024), with similar prompts. Both setups used identical training hyperparameters (see Section 5.1). As shown in the lower part of Table 2, training with our replotted charts leads to consistent performance gains, primarily due to the improved accuracy of Q/A generation enabled by explicit metadata as we have shown in Figure 2.

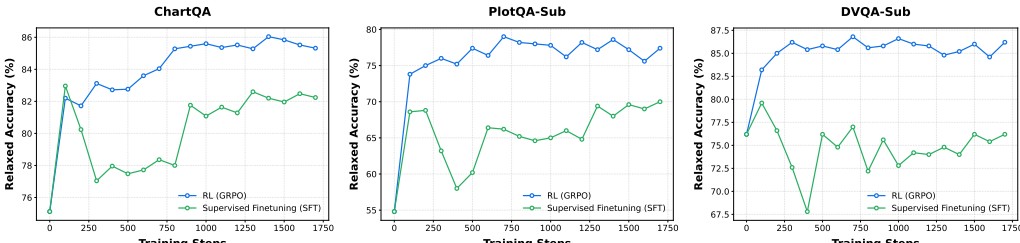

Figure 4: **Comparison between SFT and RL** (GRPO) learning curves on in-distribution benchmark (ChartQA) and out-of-distribution benchmarks (PlotQA-Sub and DVQA-Sub). The RL model shows significant improvements over the SFT model in all cases.

### 5.4 Comparing RL vs SFT for Out-of-Distribution Generalization

We compare the generalization capabilities of supervised finetuning (SFT) and reinforcement learning (RL), specifically GRPO, on out-of-distribution (OOD) tasks. To effectively evaluate OOD generalization, we exclusively train on the ChartQA (Masry et al., 2022) dataset and treat benchmarks such as *PlotQA-Sub*, and *DVQA-Sub*, and *FigureQA-Sub* as OOD test sets. Since ChartQA lacks step-by-step reasoning annotations, we leverage the Gemini Flash 2.0 (Georgiev et al., 2024) model as a teacher to generate reasoning steps. Specifically, we prompt Gemini Flash 2.0 with the chart image, question, and gold answer to obtain these reasoning steps for the SFT training dataset.

We fine-tune the *Qwen2.5-VL 3B-Instruct* model for one epoch on ChartQA using both SFT and RL approaches, with a batch size of 8 and evaluations every 100 steps. The results are presented in Figure 4. For the in-distribution ChartQA test set, both SFT and RL exhibit similar performance trends, though RL consistently shows superior performance under this small-batch training setup. More notably, we observe substantial differences between RL and SFT performances on all OOD benchmarks, clearly indicating that RL generalizes significantly better and reduces overfitting issues inherent to SFT. This also supports our decision of utilizing RL for a second stage of our training pipeline compared to existing works that only perform supervised finetuning using data generated from stronger teacher models (Masry et al., 2024a;b).

## 6 Conclusion

We introduce BIGCHARTS, a novel dataset designed to significantly enhance chart reasoning capabilities in VLMs. BIGCHARTS uniquely combines the visual authenticity of real-world chart images with the accuracy of synthetic datasets through a conditional re-plotting process. This approach addresses the longstanding issues of visual homogeneity and data estimation errors prevalent in existing datasets. We further utilize BIGCHARTS in a novel training strategy for chart reasoning, leveraging supervised finetuning and Group Relative Policy Optimization (GRPO)-based reinforcement learning with specially designed reward signals resulting in a state-of-the-art chart reasoning model, BIGCHARTS-R1. We believe that BIGCHARTS and BIGCHARTS-R1 will serve as valuable resources for fostering continued progress in the field of chart understanding and reasoning.

As future work, we plan to extend our methodology to even broader visual reasoning domains such as tables and geometric figures, and designing more diverse reward signals for tackling chart understanding tasks like chart summarization and fact-checking.

## Acknowledgments

We thank the reviewers for their valuable suggestions, which helped improve the quality of our work. We are also grateful to Ghazwa Darwiche for technical support and assistance with compute resources. This research was partially supported by the MITACS program.

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

# A    Appendix

## A.1    CoT token distribution

Figure 5 presents the distribution of the number of tokens in the CoTs generated by Gemini-Flash-2.0 for the supervised finetuning stage. More details about the dataset is provided in §3.2.

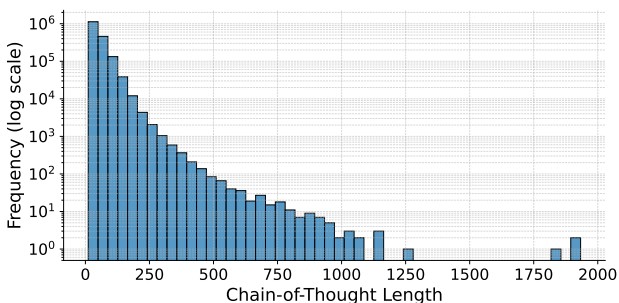

Figure 5: Distribution of the number of tokens in Chain-of-Thoughts.

## A.2    Mint-1T Filtering

We adopt a rigorous two-step filtering approach to collect chart images. In the first stage, we train a high-recall ResNet-50 (He et al., 2015) binary classifier using chart images (positive class) and natural images (negative class). We compile the chart images from the following public datasets: ChartQA (Masry et al., 2022), Chart-to-Text (Kantharaj et al., 2022b), DVQA (Kafle et al., 2018), FigureQA (Kahou et al., 2018), LRV-Chart (Liu et al., 2024a), InfoVQA (Mathew et al., 2021), AI2D (Kembhavi et al., 2016), Geo3K (Lu et al., 2021), GeoQA+ (Lu et al., 2021), Hitab (Cheng et al., 2022), robut_sqa, robut_wikisql, robut_wtq (Zhao et al., 2023), Mathqa (Amini et al., 2019), screen2words (Wang et al., 2021), visualmrc (Tanaka et al., 2021), and ureader (Ye et al., 2023). For the natural images, we use CC-12M (Changpinyo et al., 2021) and ImageNet (Russakovsky et al., 2015). We trained the classifier for one epoch and evaluated it on 100 random validation samples using a 0.95 threshold. Manual inspection showed that 88 out of 100 samples were chart-related (e.g., charts, infographics, dashboards, documents). We then applied this classifier to identify potential charts from the original image pool. In the second stage, we manually labeled 5,000 of these predicted charts to retrain the classifier for higher precision. The refined model, with a stricter threshold of 0.98, was used to filter chart images, resulting in 57,196 validated charts.

## A.3 Google Search Keywords

### List of 210 Google Search Keywords for Dataset Collection (Part 1 of 2)

- interactive choropleth map
- real-time traffic dashboard
- demographic data visualization
- election result map
- geospatial heatmap
- weather forecast dashboard
- urban planning visualization
- transportation data dashboard
- satellite imagery visualization
- historical data map visualization
- social media sentiment analysis dashboard
- engagement metrics dashboard
- customer retention dashboard
- influencer analytics dashboard
- brand reputation analysis dashboard
- advertising ROI dashboard
- content performance dashboard
- SEO dashboard
- product analytics visualization
- e-commerce sales dashboard
- patient monitoring dashboard
- public health analytics dashboard
- hospital performance dashboard
- epidemic tracking dashboard

- clinical trials visualization
- pharmaceutical data dashboard
- mental health data visualization
- vaccine distribution dashboard
- disease spread heatmap
- healthcare KPI dashboard
- student performance dashboard
- e-learning analytics dashboard
- university admissions dashboard
- course completion visualization
- learning outcomes dashboard
- education spending visualization
- teacher effectiveness dashboard
- MOOC analytics dashboard
- school district performance visualization
- global literacy rate dashboard
- public policy impact visualization
- crime statistics dashboard
- tax revenue visualization
- city budget dashboard
- employment rate dashboard
- welfare program analytics
- environmental impact dashboard
- legislation analysis dashboard
- transport infrastructure dashboard

- government transparency dashboard
- factory efficiency dashboard
- machine performance visualization
- production line analytics dashboard
- robotics monitoring dashboard
- IoT sensor data visualization
- predictive maintenance dashboard
- supply chain risk dashboard
- warehouse management visualization
- logistics data dashboard
- quality control dashboard
- stacked bar chart
- grouped bar chart
- horizontal bar chart
- 3D bar chart
- clustered bar chart
- vertical bar chart
- comparative bar chart
- segmented bar chart
- interactive bar chart
- animated bar chart

## List of 210 Google Search Keywords for Dataset Collection (Part 2 of 2)

- multiple line chart
- stacked line chart
- smoothed line chart
- trend line chart
- spline chart
- area chart
- stacked area chart
- stepped area chart
- streamgraph
- filled line chart
- exploded pie chart
- multi-level pie chart
- 3D pie chart
- semi-circle pie chart
- proportional pie chart
- nested pie chart
- doughnut chart
- half donut chart
- radial pie chart
- percentage pie chart
- colored scatter plot
- 3D scatter plot
- clustered scatter plot
- jitter plot
- regression scatter plot
- time-series scatter plot
- outlier scatter plot
- labeled scatter plot
- density scatter plot
- variable size scatter plot
- frequency histogram
- comparative histogram
- cumulative histogram
- relative frequency histogram
- overlaid histogram
- logarithmic histogram
- histogram with KDE
- animated histogram
- probability density plot
- normalized histogram
- box plot
- notched box plot
- grouped box plot
- overlapping box plot
- violin plot
- split violin plot
- half violin plot

- boxen plot
- ridge plot
- beeswarm plot
- heatmap
- correlation heatmap
- clustered heatmap
- density heatmap
- logarithmic heatmap
- calendar heatmap
- sequential heatmap
- divergent heatmap
- matrix heatmap
- geographical heatmap
- bubble chart
- radial bar chart
- polar area chart
- treemap
- sankey diagram
- chord diagram
- sunburst chart
- waterfall chart
- parallel coordinates plot
- spider chart
- candlestick chart
- OHLC chart
- funnel chart
- gantt chart
- pyramid chart
- marimekko chart
- mosaic plot
- tree diagram
- network graph
- word cloud
- time-series line chart
- stacked time-series chart
- seasonal trend decomposition chart
- moving average chart
- exponential smoothing chart
- trend decomposition chart
- forecasting chart
- growth curve chart
- rolling mean chart
- autocorrelation plot
- business infographic
- timeline infographic
- process infographic
- comparison infographic

- statistical infographic
- geographic infographic
- list infographic
- flowchart infographic
- hierarchical infographic
- pyramid infographic
- business analytics dashboard
- sales performance dashboard
- marketing analytics dashboard
- financial dashboard
- social media dashboard
- customer insights dashboard
- HR analytics dashboard
- real-time data dashboard
- project management dashboard
- executive KPI dashboard
- side-by-side bar charts
- stacked vs grouped bar chart
- multiple line charts
- comparative scatter plots
- linked histograms
- small multiples visualization
- parallel coordinate plot
- multiple pie charts
- subplots of charts
- facet grid visualization
- stock market dashboard
- cryptocurrency dashboard
- financial report visualization
- supply chain dashboard
- budget allocation dashboard
- corporate performance dashboard
- ROI visualization
- business growth dashboard
- market segmentation visualization
- economic data visualization
- genomic data visualization
- climate change visualization
- population density visualization
- energy consumption dashboard
- earthquake data visualization
- epidemiology data dashboard
- AI model performance dashboard
- scientific workflow visualization
- astronomical data visualization
- medical imaging dashboard

---

**BIGCHARTS CoT Data Generation Prompt**

```
Generate numerical and visual question-answer pairs for an LLM that we are trying to tune for Chart Numerical and Visual Reasoning.
Your response should be in a json format where each example has four fields:
input: which only asks a numerical/visual question,
chain_of_thought: a step-by-step solution that leads to the final answer,
final answer: which is the final answer to the input question based on the chart image, and question
type: the type of the question.
We have also attached the underlying code that was used to render the image so that you can have access to the underlying data and
context, however your questions should be based on the information in the chart image.
For the final answer X, follow the following instructions: * X should contain as few words as possible.
* Don't paraphrase or reformat the text you see in the image.
* If the final answer has two or more items, provide it in the list format like [1, 2].
* When asked to give a ratio, give out the decimal value like 0.25 instead of 1:4.
* When asked to give a percentage, give out the whole value like 17 instead of decimal like 0.17%
* Don't include any units in the answer.
* Try to include the full label from the graph when asked about an entity.

Generate 3 questions that contain some numerical operations such as, but not limited to, max, min, sum, average, difference,
ratio, median, mode, etc.
Generate 3 questions that not only have numerical operations,
but also some visual aspects such as leftmost, rightmost, top, bottom, middle, peak, colors, etc.
Generate 3 simple data retrieval questions that ask about values, x-labels, or legend labels from the chart.
Generate 2 yes/no numerical reasoning questions whose answers must be either Yes or No.
Generate 2 questions that ask to count some elements in the chart (e.g., the number of bars/pie slices/colors/x-labels).
Generate 1 unanswerable question which cannot be answered based on the visual information in the figure.
The answer to this question should be \Unanswerable"
Generate 1 multiple-choice question with 3, 4, 5, or more options.
The option labels can be different types: alphabet, Arabic numerals, or Roman numerals.
The answer should be the option label only.
Generate 1 conversation whose history contains 1, 2, 3 or more turns (questions and their answers) in addition to the final question
that needs to be answered based on the conversation history and the chart.
The whole conversation should in a single string in the input field.
The reasoning steps to the final question only should be in the chain_of_thought field.
The final answer to the final question should be in the final_answer field.

Remember that your questions should be based on the chart image (the code is just a helper!), and the chain_of_thought should solve
the question step by step and shows the answer in the end in this format:
<thinking> step by step here </thinking> <answer> final answer here </answer>.
```

Figure 6: Prompt to SFT data using Gemini Flash-2.0.

---

**Charts Underlying Python Code Generation Prompt**

```
Recreate the following visualization as a Python matplotlib code.
Ensure it precisely matches the original in terms of color scheme, layout, data, text elements, axis labels, title, and overall
visual appearance.
Maintain the same structure as the original image, and make sure the data is also matching it. Remember to return the python code
only.
```

Figure 7: Charts Underlying Python Code Generation Prompt.

## A.4 Prompts

We provide the prompt for generating the underlying codes of the charts in Figure 7 and Figure 8. Also, we provide the prompt used for generating CoT data in Figure 6.

## A.5 Data Analysis Prompts

We provide the prompts used for data analysis in Figure 9, 10, and 11. For topic identification, we first use the prompt in Figure 10 to generate a list of topics, and then we group the topics into 20 clusters using the prompt in Figure 11.

## A.6 Samples from BIGCHARTS

We provide sample rendered charts from our dataset in Figure 12. In addition, we also provide sample questions and step-by-step solutions in Figure 13.

**Charts Underlying React Code Generation Prompt**

```
Write a React script that replicates the provided image using the Chart.js library.
Ensure the code runs seamlessly with a simple command, such as npx babel-node file_with_code.js.
The script should render the chart and save it as chart.png. You may refer to the example below for guidance.
Remember you should return the code only! Do not return any additional text or explanation!
```

Figure 8: Charts Underlying React Code Generation Prompt.

**Chart Type Classification Prompt**

```
Analyze the provided image and identify the chart type(s) present. Return the detected chart type(s) as a JSON array of strings.
Consider the following chart types and their potential variations, including edge cases:

- **Single Line:** A line graph showing data points connected by a single line.
  Edge cases: may have markers, may be part of a larger multi-chart layout, may have a very short time series.
- **Line:** Similar to Single Line, but may imply a general line graph without specific constraints on the number of lines.
  Edge cases: same as Single Line.
- **Donut:** A circular chart with a non-solid center, showing proportions of a whole.
  Edge cases: may have very thin slices, may have text labels within the slices, may be partially obscured.
- **Stacked Bar:** A bar chart where bars are divided into segments representing subcategories, stacked on top of each other.
  Edge cases: may have very small segments, may be horizontal, may have overlapping labels.
- **Line + Bar:** A combination chart with both line and bar elements.
  Edge cases: may have multiple lines and bars, may have different scales for lines and bars, may have overlapping elements.
- **Area Chart:** A line chart with the area between the line and the axis filled in.
  Edge cases: may have overlapping areas, may have a gradient fill, may be partially transparent.
- **Scatterplot:** A chart showing data points as individual markers on a coordinate plane.
  Edge cases: may have overlapping points, may have different marker sizes or colors, may have trend lines.
- **Group Bar:** A bar chart where bars are grouped by category, showing multiple values for each category.
  Edge cases: may have very small groups, may have overlapping labels, may be horizontal.
- **Pie:** A circular chart divided into slices representing proportions of a whole.
  Edge cases: may have very thin slices, may have text labels within the slices, may be partially obscured.
- **Bar:** A chart with rectangular bars representing data values.
  Edge cases: may be horizontal, may have overlapping labels, may have varying bar widths.
- **Multi Line:** A line graph showing multiple lines representing different data series.
  Edge cases: may have a large number of lines, may have overlapping lines, may have different line styles.
- **Multi Bar:** A bar chart showing multiple bars representing different data series.
  Edge cases: may have a large number of bars, may have overlapping labels, may be horizontal.
- **Histogram:** A chart with vertical bars representing the frequency of data values in a range.
  Edge cases: may have overlapping bars, may have a gradient fill, may be horizontal.
- **Boxplot:** A chart with a box and whiskers representing the distribution of data values.
  Edge cases: may have overlapping whiskers, may have a gradient fill, may be horizontal.
- **Heatmap:** A chart with a grid of cells representing the values of a matrix.
  Edge cases: may have overlapping cells, may have a gradient fill, may be horizontal.
- **Violin Plot:** A chart with a violin shape representing the distribution of data values.
  Edge cases: may have overlapping violins, may have a gradient fill, may be horizontal.

If the chart type is ambiguous or not explicitly listed above, return 'Other'.
- If multiple chart types are present, include all of them in the array.
- If the chart is not one of the above, report the most appropriate chart type for it.
- If no chart type can be identified, return an empty array: `[]`.

Output format:
```json
{
    "chart_types": ["Single Line", "Pie"]
}
```
```

Figure 9: The prompt used for identifying the chart type.

**Chart Topic Identification Prompt**

```
Analyze the provided chart image and report the topic of the information it represents.
Focus on the following aspects to determine the chart's core topic:
 - Data Types: What kind of data is being represented (e.g., numerical, categorical, temporal)?
 - Variables: Identify the key variables shown in the chart (e.g., axes labels, legend entries).
 - Relationships: How do the variables relate to each other? What patterns or trends are being shown?
 - Chart Type: What type of chart is it (e.g., bar chart, line graph, pie chart)?
   This often provides clues about the purpose of the data presentation.
 - Contextual Clues: Are there any titles, labels, or annotations within the chart that provide direct information about the topic?
 - Image Filename: The image filename is: "{image_filename}". This may provide additional context about the chart's content
   and source.

If the image filename is not meaningful, ignore it.
The main source of information is the title of the chart and the axis labels.

Based on your analysis, provide a concise and precise description of the chart's topic in 3 words or fewer, and the subtopic in
3 words or fewer. Focus on the core subject matter being presented, not the specific data values.

Always return an array of topics.
If there are multiple topics, return all of them in an array.
If there is no clear topic, return an empty array "[]".

Output format:
```json
{
    "topic main": ["Topic1", "Topic2", "Topic3"]
    "topic sub": ["Topic4", "Topic5", "Topic6"]
}
```
```

Figure 10: The prompt used for identifying the chart topics.

**Chart Topic Clustering Prompt**

```
You are an expert in clustering and topic analysis. You are given a list of topics and their subtopics. You must cluster them into
20 main clusters. Your response must be in the form of a JSON object, the key is the cluster name, and the value is a list of topics
and subtopics from the input list. Make sure to put all the given topics in at least one cluster; the number of topics in each
cluster is important to me.

Output format: {cluster name: [{'main': main topics in this cluster, 'sub': sub topics in this cluster}]}
```

Figure 11: The prompt used for clustering the chart topics.

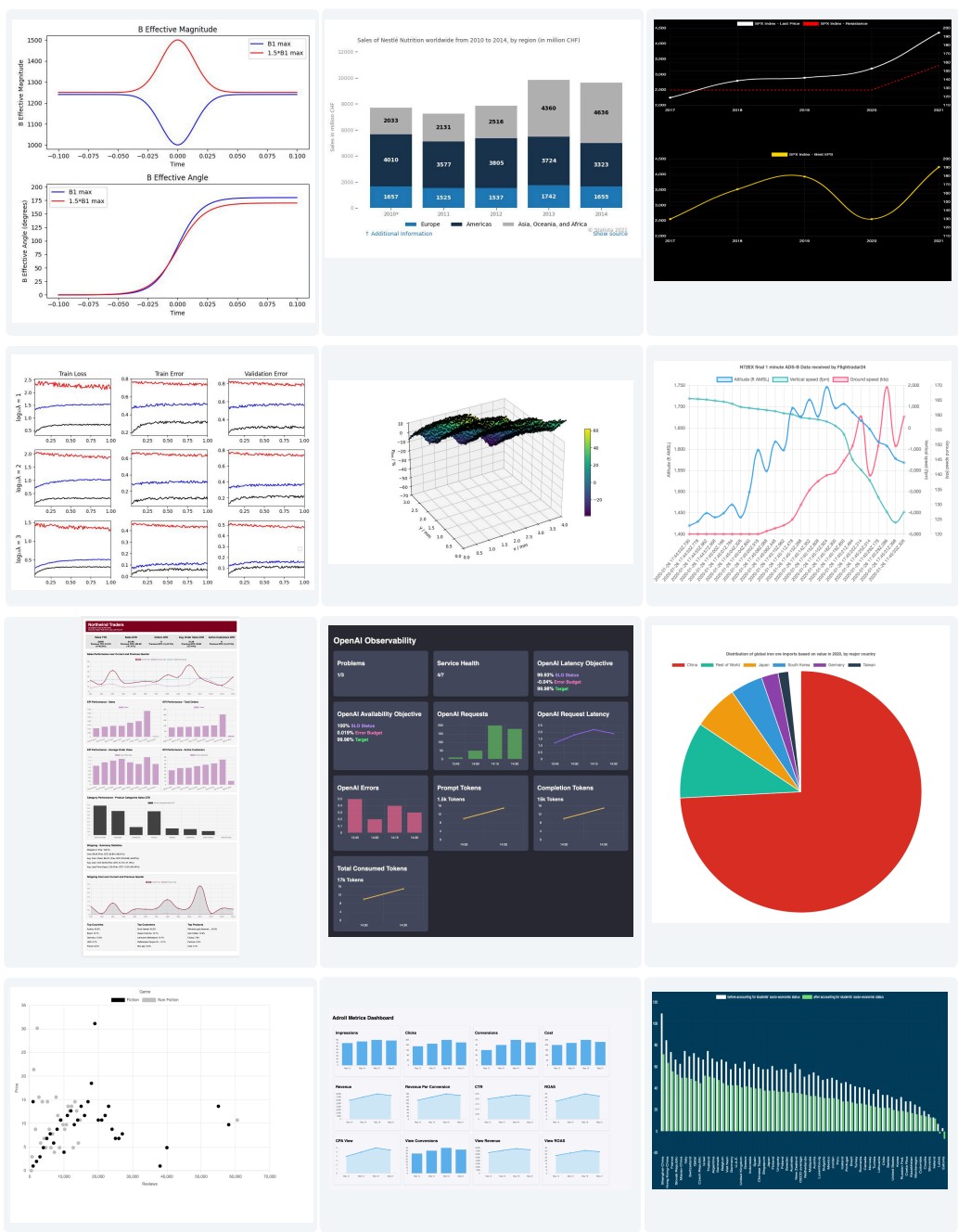

Figure 12: Examples of different rendered charts from our BIGCHARTS dataset.

## A.7 Sample Outputs from BigCharts-R1 Model

Sample Predictions from BIGCHARTS-R1 on the ChartQA benchmark are presented in Figure 14. Sample Predictions from BIGCHARTS-R1 on the PlotQA benchmark are presented in Figure 14. Sample Predictions from BIGCHARTS-R1 on the DVQA benchmark are presented in Figure 16. Sample Predictions from BIGCHARTS-R1 on the Figureqa benchmark are presented in Figure 17. Sample Predictions from BIGCHARTS-R1 on the Figureqa benchmark are presented in Figure 18.

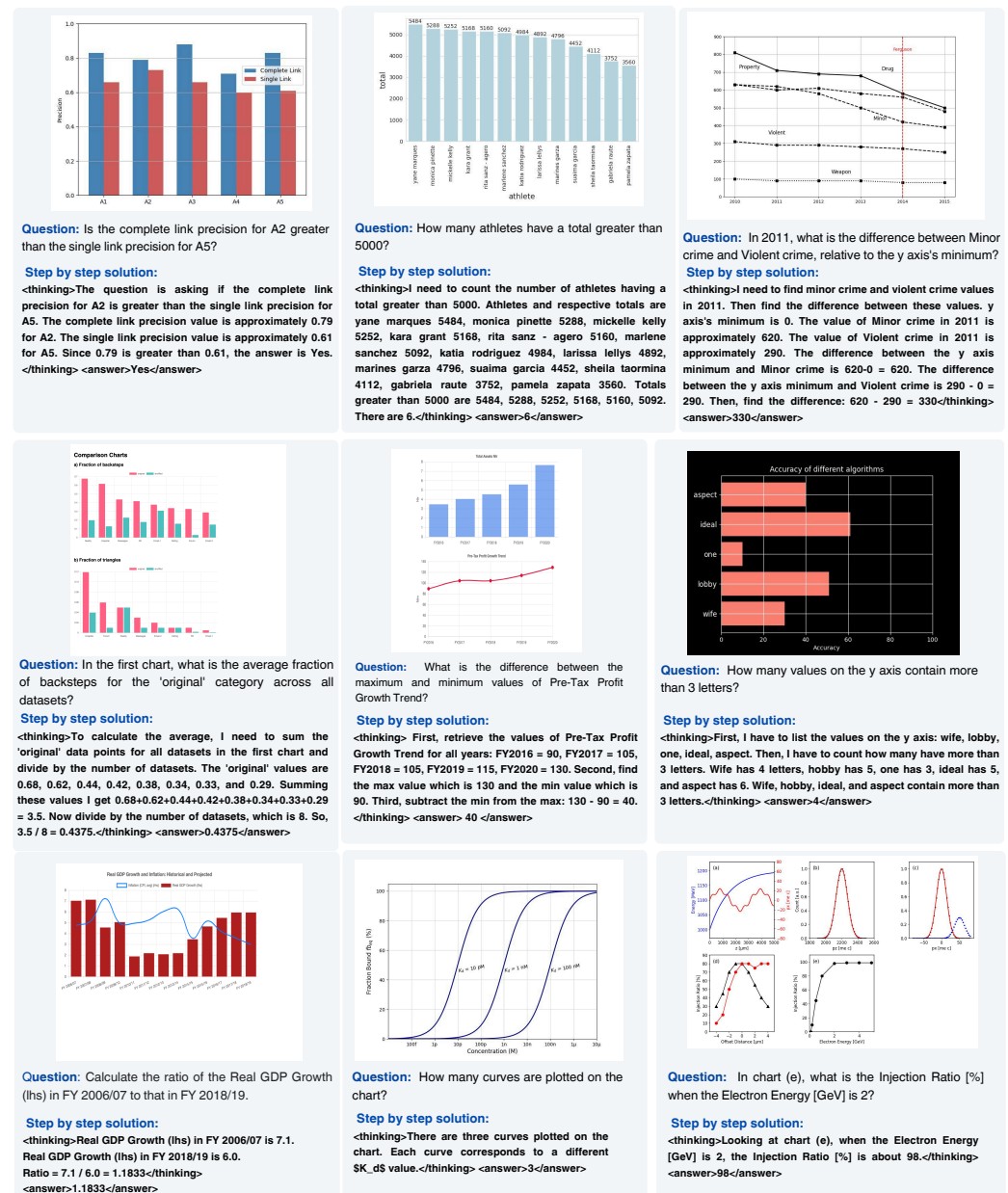

Figure 13: Examples of different questions and their step-by-step solutions from our BIGCHARTS dataset.

## A.8 Charts Replotting

Examples of original charts and their replotted variants in BIGCHARTS in Figure 19.

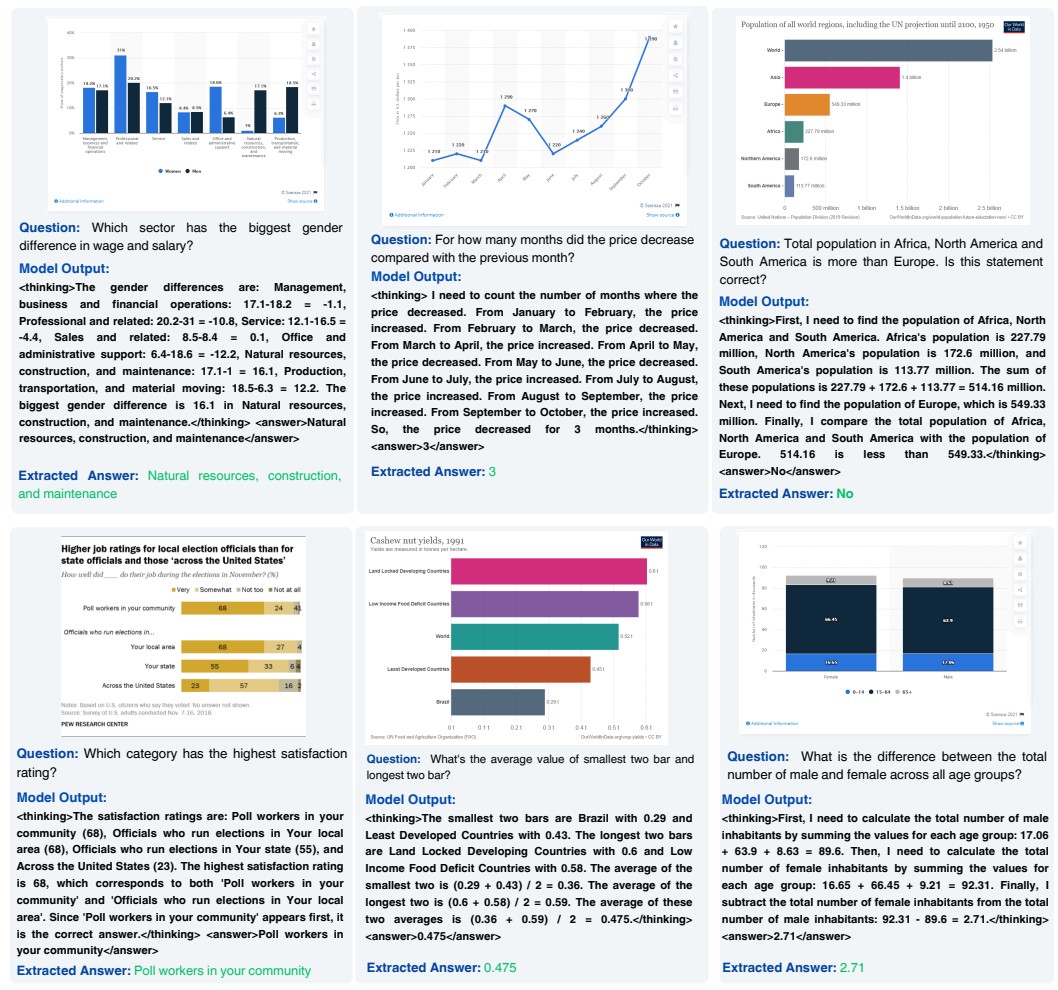

Figure 14: Examples of different outputs from BIGCHARTS-R1 on the ChartQA benchmark.

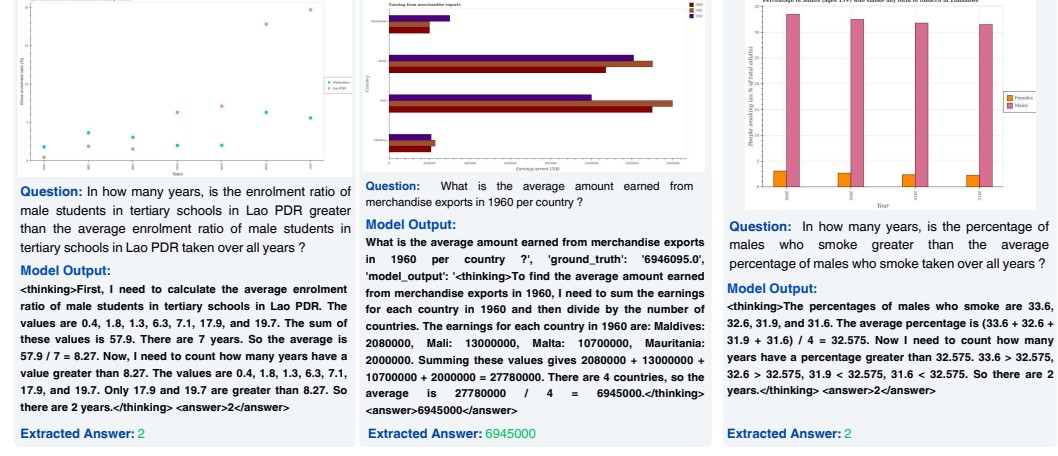

Figure 15: Examples of different outputs from BIGCHARTS-R1 on the PlotQA benchmark.

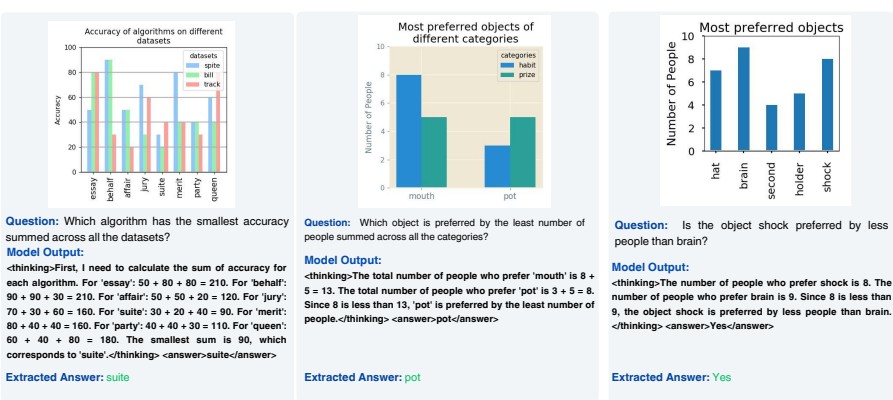

Figure 16: Examples of different outputs from BIGCHARTS-R1 on the DVQA benchmark.

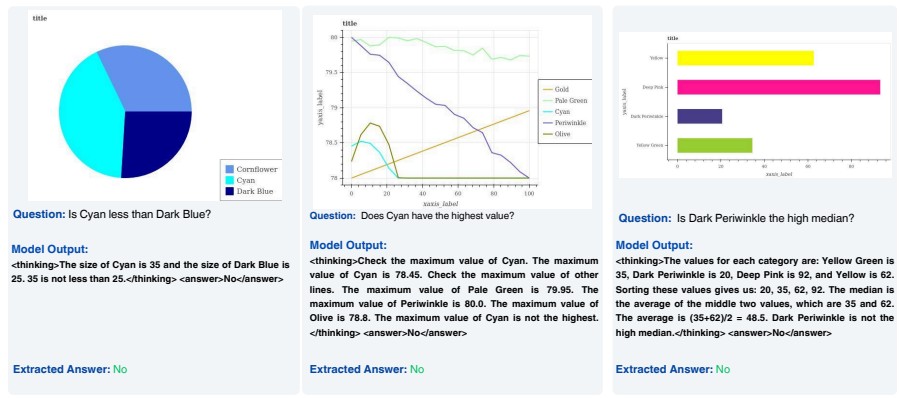

Figure 17: Examples of different outputs from BIGCHARTS-R1 on the Figureqa benchmark.

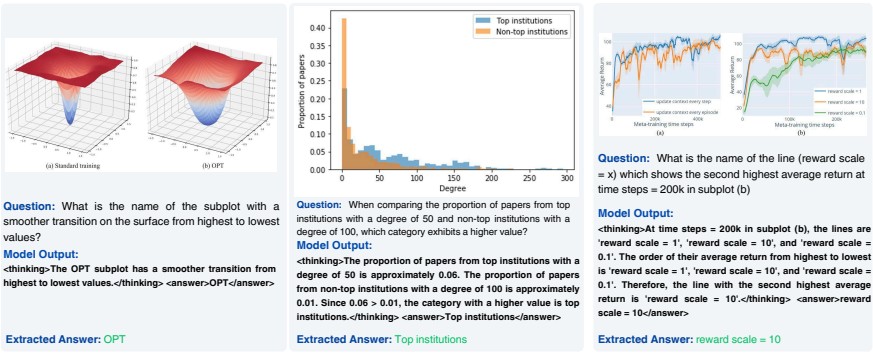

Figure 18: Examples of different outputs from BIGCHARTS-R1 on the CharXiv benchmark.

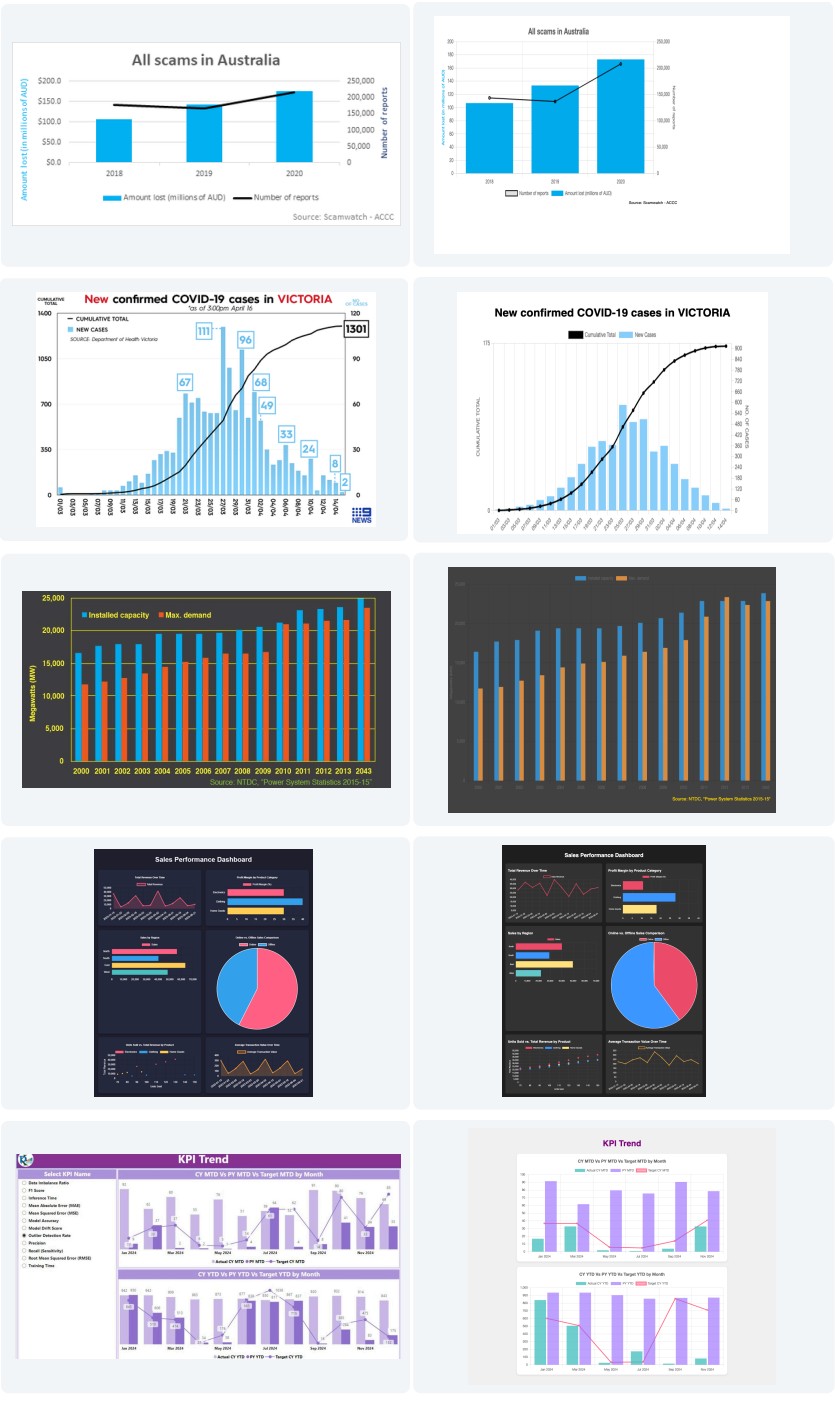

Figure 19: Examples of original charts (the left) and their replotted variants (the right) in BIGCHARTS.

