# OpenReview forum: "BigCharts-R1: Enhanced Chart Reasoning with Visual Reinforcement Finetuning"
_colmweb.org/COLM/2025/Conference — COLM 2025_

### Official Review · Reviewer_8ZTe · 2025-04-22

**Rating:** 6
**Confidence:** 5
**Ethics Flag:** 1

**Summary:**

The paper introduces BIGCHARTS-R1, a state-of-the-art framework for chart reasoning in vision-language models (VLMs). It addresses key limitations in existing models—namely, low visual diversity, reliance on inaccurate chart data, and overfitting from supervised fine-tuning alone—by proposing two main innovations: (1) BIGCHARTS, a visually diverse and accurate chart dataset curated through a novel replotting process that combines real-world chart images with synthesized data; and (2) a two-stage training approach combining supervised fine-tuning (SFT) with Group Relative Policy Optimization (GRPO)-based reinforcement learning. Extensive experiments across synthetic and real-world benchmarks demonstrate that BIGCHARTS-R1 outperforms both open-source and closed-source models in chart question answering and generalization tasks.

**Reasons To Accept:**

1. The authors propose a novel dataset curation strategy that combines real-world chart visual diversity with accurate underlying data through a replotting process. This addresses the common issue of synthetic chart homogeneity and real-chart data inaccuracy, leading to a more robust and authentic training dataset.

2. By integrating supervised fine-tuning with Group Relative Policy Optimization (GRPO)-based reinforcement learning, the paper improves both chart comprehension and reasoning abilities. The use of verifiable, task-specific reward functions (e.g., Chart Error Rate Reward) further strengthens the model’s performance and generalization.

3. The model, BIGCHARTS-R1, achieves state-of-the-art results across multiple chart QA benchmarks, outperforming existing open-source and closed-source models, including larger ones. Its performance gain on out-of-distribution datasets underscores the robustness and effectiveness of the proposed methodology.

**Reasons To Reject:**

1. While the paper focuses heavily on chart question answering, it provides limited exploration of other important chart-related tasks like summarization, captioning, or fact-checking. This narrows the scope of its generalizability across diverse chart understanding needs.

2. The data generation pipeline relies on vision-language models (e.g., Gemini Flash 2.0) to generate plotting code and question-answer pairs. Although followed by re-rendering and filtering, this introduces a risk of propagating model biases or subtle inaccuracies in both visual and semantic content.

3. The proposed training pipeline, especially the reinforcement learning stage using GRPO with verifiable rewards, demands substantial computational resources (e.g., 8× H100 GPUs). This might limit reproducibility and accessibility for smaller labs or practitioners with limited hardware.

---

> ### Author Response · Authors · 2025-06-03
>
> Thank you for your constructive comments. We address your concerns below:
>
> # Concern 1: Evaluation on other Chart Tasks
> Our primary focus is on the chart question answering task due to the availability of multiple benchmarks emphasizing mathematical and visual reasoning, such as ChartQA, CharXiv, FigureQA, DVQA, and PlotQA. These benchmarks align well with our aim of enhancing reasoning capabilities. On the other hand, existing chart summarization/captioning benchmarks (e.g., Chart-to-Text [1]) primarily measure descriptive abilities using BLEU scores which offer limited insights into reasoning performance. We acknowledge that chart fact-checking is another valuable task for evaluating reasoning. Hence,  we conducted additional evaluations on the ChartCheck benchmark [2]. As shown in the table below, our models (BigCharts-R1-3B and BigCharts-R1-7B) achieve higher accuracies than all open-source models on this task. Compared to closed-source models, our models perform competitively: although slightly behind GPT-4o and Gemini Flash 2.0 on Test 1, BigCharts-R1-7B surpasses them on Test 2. This highlights the strong out-of-distribution generalization of our models on diverse reasoning-intensive chart tasks, even though our BigCharts dataset was not explicitly designed for fact-checking.
>
> | Model | ChartCheck Test 1 | ChartCheck Test 2 |
> |--|--|--|
> | GPT4o| 84.49| 82.36 |
> | Claude 3.5 Sonnet| 57.96 | 51.88|
> | Gemini Flash 2.0| 84.91| 84.19|
> | InternVL2_5-1B | 61.92 | 61.67|
> | InternVL2_5-2B| 69.19| 69.52|
> | Phi-3.5-Vision| 74.22| 72.37|
> | InternVL2_5-8B | 77.96| 77.67 |
> | llava_next_mistral_7b| 61.17| 62.99|
> | Llama-3.2-11B-Vision-Instruct  | 0.10| 0.10|
> | ChartGemma-3B | 71.50 | 74.31|
> | Qwen2.5-VL-3B (CoT) | 47.06| 55.45 |
> | Qwen2.5-VL-3B + SFT| 77.75| 81.04|
> | **BigCharts-R1-3B** | **78.93**| **83.08**|
> | Qwen2.5-VL-7B (CoT)| 71.12 | 71.15 |
> | Qwen2.5-VL-7B + SFT| 80.96| 83.28|
> | **BigCharts-R1-7B** | **84.28** | **84.30**|
>
> # Concern 2: Risk of Model Bias, Inaccuracies of the Gemini Flash 2.0 model.
> We acknowledge that relying on LLMs for synthetic data generation introduces certain limitations and potential inaccuracies. Nonetheless, synthetic generation remains essential for constructing large-scale datasets efficiently, given the high cost of human annotation.
>
> Our novel conditional replotting approach helps to mitigate these risks compared to previous methods. By replotting charts and providing the vision-language model (Gemini Flash 2.0) with both accurate metadata and visually authentic re-rendered images, our method enhances the accuracy of the resulting question-answer pairs. In contrast, previous chart data generation methods (e.g., ChartGemma and TinyChart) generate Q/A pairs using only the chart image, which can lead to greater inaccuracies due to the LLM’s need to estimate data values directly from the image.
>
>
> To empirically validate this advantage, we compared the performance of Qwen2.5-VL-3B trained with Q/A pairs generated directly from original charts against those generated using our replotting method. Both models are trained using the same hyperparameters reported in Section 5.1 to ensure a fair comparison. As shown below, our approach consistently achieves higher performance, largely due to the improved accuracy of the generated Q/A pairs.
>
> | Method | FigureQA Val 1 | FigureQA Val 2 | DVQA Easy | DVQA Hard | PlotQA T1 | PlotQA T2 | ChartQA Avg. | CharXiv Reas. | CharXiv Desc. | Avg.      |
> | ---|---|---|---|--|--|--| --| --|-- |-- |
> | Original Charts  | 73.70 | 73.10 | 75.20| 72.5 | 72.40| 52.40 | 82.24 | 30.90| **63.62** | 66.22  |
> | Replotted Charts | **76.10** | **75.70** | **76.30** | **73.80** | **74.60** | **58.40** | **84.60** | **36.00** | 62.85 | **68.71** |
>
> We believe our conditional replotting method presents a promising solution not only for chart reasoning but also for generating more accurate and diverse datasets in other visual reasoning tasks, such as tables and geometric figure comprehension. We also used Gemini Flash 2.0 due to its affordability; teams with greater resources may wish to explore larger, more expensive, and potentially more accurate models such as Claude 3.5 Sonnet with our novel replotting approach.
>
> # Concern 3: Demand for Computational Resources
>
> We acknowledge that our experiments required relatively large computational resources (e.g., 8×H100 GPUs), which is increasingly common given the size of modern vision-language models, even though we used the smallest Qwen2.5-VL models (3B and 7B). Researchers with limited resources can fine-tune smaller models on our dataset and leverage our reinforcement learning approach with more modest hardware.
>
> Thank you for your thoughtful review. We hope our response has addressed your concerns and **kindly ask you to consider raising your score** to support the dissemination of our work within the ML community.
>
> [1] Chart-to-Text: https://arxiv.org/abs/2203.06486
>
> [2] ChartCheck: https://arxiv.org/abs/2311.07453

---

> > ### Author Response · Authors · 2025-06-08
> >
> > Dear Reviewer,
> >
> > Thank you for dedicating your time and effort to reviewing our manuscript and for providing insightful suggestions. As the author-reviewer discussion phase nears its conclusion, we would like to ensure that our responses have adequately addressed your concerns. Please feel free to reach out if you have any additional questions or require further clarification.
> >
> > Best Regards,
> >
> > Authors

---

### Official Review · Reviewer_P2Yy · 2025-05-12

**Rating:** 6
**Confidence:** 4
**Ethics Flag:** 1

**Summary:**

This is a benchmarking paper. This work introduces a dataset for the chart question-answering task, incorporating existing synthetic datasets and real-world chart data, and a training framework integrating supervised fine-tuning with Group Relative Policy Optimization. The experiment results show improvements using their proposed training framework.

**Questions To Authors:**

Q1: For the replotting process, what if the replotted chart mismatches the original chart? How to check whether the plot is reproduced by the generated code, including style and content? Figure 19 shows the figures that are mismatched.

Q2: For Common Crawl processing(Line 138-143), where are labeled chart images from in the first stage? What’s the accuracy after training in the first stage? You have labeled chart images in the first stage and also manually labeled data in the second stage, so why not directly label 5000 charts of data for training a classifier? Furthermore, if you have labeled chart images, why not directly using in-context learning with LLM? The binary classification of identifying whether it is a chart image or not is quite a simple task.

Q3: As Table 3 shows, the BigCharts is an imbalanced dataset regarding chart types, chart topics, question types and answer types. I think it should be a similar situation in the baseline benchmarks, so the sole exact accuracy metric may not be suitable for the evaluation. The additional metrics might be required.

Q4: Do the sampled baseline benchmarks follow the same distribution as their original benchmark?

Q5: It might require additional studies to show whether BigCharts solves the limitations of existing benchmarks, i.e., visual homogeneity of the dataset, estimation errors in underlying chart data(Line 51-52). Suggesting that (1)evaluating Closed/Open-source Models on the BigCharts dataset and (2) training Chart-Specific LVLMs with the BigCharts and then evaluating them on their dataset. See the performance changes in these two studies.

**Reasons To Accept:**

- This work introduces a large dataset for the chart question-answering task, incorporating existing synthetic datasets and real-world charts data.
- The proposed training framework shows the improvements in the in-domain and out-of-domain datasets.
- The writing is overall clear and easy to follow.

**Reasons To Reject:**

C1: The technical contribution is limited. The training framework is a combination of technologies in other works, and the proposed Chart Error Rate Reward is impractical for non-numeric answers, i.e., exact-matching criterion.

C2: It may need more studies to support that their proposed BigCharts dataset solves the limitations of existing benchmarks.

C3: It still may need some clarification.

---

> ### Author Response · Authors · 2025-06-03
>
> Thank you for your constructive comments. We address your concerns below:
>
> # Concern 1: Limited Technical Contribution
> We would like to clarify that the main contribution of our paper is the **BigCharts dataset**, created through a novel data generation pipeline (conditional chart replotting) designed to address key limitations of existing methods, as illustrated in Figures 1 and 2 (see lines 75–81), such as visual homogeneity and estimation errors. While our training framework largely builds on established techniques, our focus is on dataset creation pipelines rather than new model architectures. Our experiments demonstrate that models fine-tuned on BigCharts consistently outperform those trained on prior datasets for chart question answering such as TinyChart and ChartGemma models. Additionally, to the best of our knowledge, we are the first to apply the GRPO reinforcement learning algorithm to the chart reasoning and question answering domain, and our results highlight its effectiveness in enhancing visual mathematical reasoning for charts.
>
> # Concern 2: More studies on BigCharts
> Please refer to our responses to the questions below (Question 5)
>
> # Concern 3: More Clarification
> Please refer to our responses to the questions below (Question 5)
>
> # Question 1: Mismatches in the Replotting Process
> Our replotting approach is designed to approximate the visual style and content of the original chart, rather than achieve a perfect match. Replicating charts exactly is extremely challenging, even for advanced LLMs like Gemini, due to unavoidable data estimation errors. Our goal is to generate a similar chart that preserves diversity in both visuals and content; as a result, we use Gemini to produce code for a closely related chart with estimated data values, and subsequently discard the original image.
>
> If current LLMs such as Gemini were capable of perfectly reproducing original charts and estimating all data values accurately, we could generate precise Q/A pairs directly from the chart images without the need for replotting. However, since perfect accuracy cannot be ensured, our replotting process allows us to generate similar chart images that maintain visual diversity while providing reliable, high-quality metadata (i.e., the underlying code).
>
> To assess whether our conditional chart replotting process maintains visual and topical diversity, we analyzed the distributions of chart types and topics (as presented in Figure 3 and Section 3.2) by prompting Gemini Flash 2.0 to classify both the replotted and original charts. As shown in the table below, the distributions of chart types and topics are very similar. This demonstrates that our conditional replotting approach, despite its imperfections, effectively preserves the diversity of visual styles and content in the generated dataset, BigCharts.
>
> | Chart Types  | Replotted Charts (%) | Original Charts (%) |
> |---|---|---|
> | Bar  | 26.8 | 24.8 |
> | Multi Bar  | 7.9  | 7.8 |
> | Stacked Bar | 3.1   | 4.4  |
> | Line  | 23.1 | 17.4  |
> | Multi Line  | 22.2 | 24.4  |
> | Scatter Plot  | 7.2  | 6.4 |
> | Area Chart  | 3.0 | 3.1 |
> | Pie Chart  | 2.1 | 2.8  |
> | Donut Chart | 2.0 | 2.4 |
> | Line & Bar  | 1.8 | 1.9 |
> | Others  | 0.8  | 4.6 |
>
> | Chart Topics  | Replotted Charts (%) | Original Charts (%) |
> |---|---|---|
> | Finance & Economy  | 39.0 | 31.6 |
> | Data Science | 27.5  | 30.4  |
> | STEM - Physics  | 13.5 | 11.6   |
> | STEM - Chemistry   | 8.7 | 2.6 |
> | STEM - Biology   | 4.7  | 5.6 |
> | STEM - Mathematics | 2.8 | 4.5 |
> | Time Series  | 3.7 | 6.6 |
> | Others  | 0.2  | 7.2 |
>
> # Question 2: Filtering process for Common Crawl
> Due to space constraints, we originally placed the details of the filtering process in Appendix A.2 (lines 611–627), where we list all datasets used for the first-stage classifier training—both positive (ChartQA, Chart-to-Text, AI2D, LRV-Chart, etc.) and negative examples (CC-12M, ImageNet). These datasets were chosen for their diversity and because they could be compiled without manual annotation. The first-stage classifier achieved 88% precision on a small validation set. To further improve precision, we manually annotated 5,000 challenging samples, primarily false positives, identified by the model and used them to finetune the classifier further. This raised the classifier’s precision to 97%.
> We opted not to use in-context learning with LLMs due to the high cost and latency associated with processing the massive scale of the Mint-1T dataset split (5.07 million samples). A ResNet-50 classifier provided a significantly more efficient and cost-effective solution for large-scale filtering. We will clarify these points in the main text in the revised manuscript.

---

> > ### Author Response · Authors · 2025-06-03
> >
> > # Question 3: Imbalance dataset
> > We agree that, as shown in Figure 3, BigCharts is not perfectly balanced across chart types, chart topics, question types, and answer types. This distribution is intentional, as it more closely mirrors the natural occurrence of these elements in real-world data. For instance, bar and line charts are far more common in practice, which is reflected in their higher representation within the dataset, while less common types (e.g., area charts, scatter plots) appear less frequently. Similarly, questions about visual features are more prevalent because users typically focus on such aspects when analyzing charts [1, 2]. Enforcing a uniform representation across all types would create an artificial distribution that does not reflect real-world scenarios.
> >
> > Regarding evaluation, we used exact and relaxed accuracy metrics because they are the official metrics adopted by the authors of each benchmark, FigureQA, DVQA, PlotQA, ChartQA, and CharXiv. This ensures consistency and enables direct comparison with prior work.
> >
> > # Question 4: Benchmarks Distributions
> > Yes, we randomly sampled 1000 examples from each benchmark using independent uniform sampling to ensure that the sampled subsets maintain the same distribution as the original benchmarks.
> >
> > # Question 5: Additional Studies of BigCharts vs Existing Datasets.
> > We agree that further studies could provide additional evidence regarding the benefits of BigCharts over existing benchmarks. We would like to clarify that BigCharts is designed as a training dataset, similar to prior works such as ChartGemma and TinyChart, not as an evaluation benchmark. As shown in Table 1, fine-tuning our model, Qwen2.5-VL-3B, on BigCharts leads to better performance compared to models fine-tuned on previous datasets such as TinyChart and ChartGemma.
> >
> > We also recognize the importance of conducting rigorous and fairer comparisons that eliminate the bias introduced by different backbone models (Qwen2.5-VL, PaliGemma, TinyLLaVa). To address this concern, we fine-tuned the same backbone model (Qwen2.5-VL-3B) on the publicly available ChartGemma and TinyChart datasets. For evaluation, we utilized the codebase and "program-of-thought" answer formats recommended by the datasets authors to ensure optimal and fair performance. The results in the table below confirm that fine-tuning with our BigCharts dataset consistently outperforms these datasets, particularly on challenging benchmarks such as ChartQA, PlotQA, and CharXiv.
> >
> > | Method    | FigureQA Val 1 | FigureQA Val 2 | DVQA Val Easy | DVQA Val Hard | PlotQA T1 | PlotQA T2 | ChartQA Avg. | CharXiv Reas. | CharXiv Desc. | Avg.   |
> > |---|---|---|---|---|---|---|---|---|---|---|
> > | Qwen2.5-VL-3B (ChartGemma)    | 78.30| 79.10  | 73.30| 74.30   | 40.50     | 58.60     | 68.48        | 10.70         | 39.25            | 58.05  |
> > | Qwen2.5-VL-3B (TinyChart)     | 47.30  | 48.80  | 51.40 | 48.19         | 57.09     | 50.00   | 70.28   | 24.60   | 41.62            | 48.80  |
> > | **Qwen2.5-VL-3B (BigCharts)**     | 76.10          | 75.70          | 76.30         | 73.80         | 74.60     | 58.40     | 84.60        | 36.00         | 62.85            | **68.70**  |
> >
> > Furthermore, we conducted additional experiments comparing the performance of Qwen2.5-VL-3B when trained with Q/A pairs generated directly from the original (real-world) charts versus our synthetic replotting approach, which provides both the replotted chart image and the underlying data. For consistency, Q/A pairs for the original charts were generated using Gemini Flash 2.0 with prompts similar to those used for the replotted charts (see Figure 6 in the Appendix). Both models were trained with identical hyperparameters as described in Section 5.1 to ensure a fair comparison. Results in the table below demonstrate that our replotted charts approach leads to improved performance, primarily due to more accurate Q/A generation enabled by metadata availability.
> >
> > | Method           | FigureQA Val 1 | FigureQA Val 2 | DVQA Easy | DVQA Hard | PlotQA T1 | PlotQA T2 | ChartQA Avg. | ChartarXiv Reas. | ChartarXiv Desc. | Avg.  |
> > |------------------|----------------|----------------|-----------|-----------|-----------|-----------|--------------|------------------|------------------|-------|
> > | Qwen-2.5-VL-3B (Original Charts)  | 73.70         | 73.10          | 75.20     | 72.50     | 72.40     | 52.40     | 82.24        | 30.90            | **63.62** | 66.22 |
> > | Qwen-2.5-VL-3B (Replotted Charts) | **76.10**  | **75.70**  | **76.30** | **73.80** | **74.60**  | **58.40** | **84.60** | **36.00** | 62.85   | **68.71** |
> >
> > Thank you for your thoughtful review. We hope our response has addressed your concerns and **kindly ask you to consider raising your score** to support the dissemination of our work within the ML community.
> >
> > [1] ChartQA: https://arxiv.org/abs/2203.10244
> >
> > [2] CharXiv: https://arxiv.org/abs/2406.18521

---

> > > ### Author Response · Authors · 2025-06-08
> > >
> > > Dear Reviewer,
> > >
> > > Thank you for dedicating your time and effort to reviewing our manuscript and for providing insightful suggestions. As the author-reviewer discussion phase nears its conclusion, we would like to ensure that our responses have adequately addressed your concerns. Please feel free to reach out if you have any additional questions or require further clarification.
> > >
> > > Best Regards,
> > > Authors

---

> > > ### Comment · Reviewer_P2Yy · 2025-06-10
> > >
> > > Thank you for the authors' detailed response and for providing additional experiment results! I have read your response carefully, and I think your additional information addressed my main concerns about limited technical contribution and requiring additional studies. Therefore, I will raise my score.
> > >
> > > Just one more question, is there a better reward method for non-numeric answers (current is exact-matching criterion) in the Chart Error Rate Reward?

---

> > > > ### Author Response · Authors · 2025-06-10
> > > >
> > > > Thank you so much for following up and for updating your score. Regarding your question about alternatives to the exact match criterion for non-numeric answers, we believe that Levenshtein distance could serve as a useful continuous metric. This approach would reward the model for minimizing the distance between the predicted and ground truth answers. For example, we can use a normalized score such as 1 - Levenshtein_distance / (len1 + len2), where len1 and len2 are the character lengths of the ground truth and predicted answers, keeping the score in the [0, 1] range.
> > > >
> > > > Unfortunately, due to the limited time remaining in the discussion period, we won't be able to get the results in time. However, we will make sure to address this experiment in the revised manuscript.

---

### Official Review · Reviewer_Lpvk · 2025-05-13

**Rating:** 5
**Confidence:** 4
**Ethics Flag:** 1

**Summary:**

This paper addresses the challenge of poor generalization in chart reasoning by current vision-language models (VLMs), largely due to limited visual diversity and inaccurate chart data in training sets. To overcome this, the authors introduce BIGCHARTS, a dataset pipeline that re-generates real-world chart images with accurate underlying data, and BIGCHARTS-R1, a model trained using supervised finetuning and GRPO-based reinforcement learning with chart-specific rewards. Experiments on five benchmarks demonstrate that BIGCHARTS-R1 achieves state-of-the-art performance and significantly improves out-of-distribution generalization.

**Questions To Authors:**

See "Reasons To Reject"

**Reasons To Accept:**

The paper presents a moderately new approach by combining real-world chart images with accurate data through a conditional replotting process.  The integration of supervised finetuning and GRPO-based reinforcement learning is appropriate for addressing generalization issues in chart reasoning.  The methodology is clearly described, with a well-structured dataset pipeline and targeted reward design. Empirical results on multiple benchmarks support the effectiveness of the approach, and the framework shows a few potential applicability to related tasks in visual-language understanding.

**Reasons To Reject:**

1. **Justification of GRPO**: The choice of GRPO as the reinforcement learning algorithm is not sufficiently justified. The paper does not compare GRPO with other standard RL methods (e.g., PPO), leaving unclear whether GRPO is necessary or advantageous for this task.

2. **Experimental Instability and Setup Issues**: In Section 5.3, the paper claims stable performance for SFT on in-distribution data, yet Figure 4 shows a consistent early peak followed by a sharp drop across multiple datasets. This pattern suggests potential issues in the experimental setup, such as small batch sizes or unshuffled training data, which are not discussed.

3. **Limited Analysis of Reward Function Design**:  The proposed CERM is a central component, but the paper provides little analysis or ablation to assess its specific contribution. Moreover, the final reward is a simple sum of two components (Eq. 5), with no discussion of other possible combination strategies or weighting schemes.

4. **Insufficient Dataset Construction Details**:  The dataset collection process, particularly from heterogeneous sources such as ChartGemma, FigureQA, and Common Crawl, lacks detail regarding quality control, filtering, and normalization. This omission hinders reproducibility and raises concerns about dataset consistency.

5. **Presentation and Structural Issues**: The paper has presentation inconsistencies, such as unnecessary sub-subsections (e.g., only 4.2.1 under 4.2) and inconsistent abbreviation usage (e.g., "reinforcement learning", "Group Relative Policy Optimization" vs. "RL", "GRPO") in the paper. These issues affect clarity and readability.

---

> ### Author Response · Authors · 2025-06-03
>
> Thank you for your constructive comments. We address your concerns below:
>
> # Concern 1: Justification of GRPO
> We use GRPO for its computational and memory efficiency compared to PPO.  As noted in lines 240-243, GRPO eliminates the need for a separate value model, typically as large as the policy model, by estimating the baseline directly from the scores of multiple sampled outputs [1].  Given the large size of the models used in our experiments (3B and 7B parameters), GRPO provides a practical and efficient solution under limited compute and memory resources.
>
> # Concern 2: Experimental Instability
> We would like to clarify that we randomly shuffle the training dataset before fine-tuning to avoid any bias or inconsistency related to data order. We agree that the early peak and subsequent drop observed in the SFT curves in Figure 4 are primarily due to the small batch size (8) used in these experiments. We intentionally used this small batch size in SFT to match the batch size used in RL training for this particular ablation study in Section 5.3, allowing for a more direct and fair step-by-step comparison in Figure 4. Given the large size of our models (Qwen2.5-VL-3B and Qwen2.5-VL-7B), larger batch sizes were not feasible for RL due to hardware constraints. Nevertheless, RL training with the same small batch size produces stable learning curves and steadily improving validation accuracy, as shown in Figure 4, particularly on out-of-distribution benchmarks such as PlotQA and DVQA. This shows that our GRPO training is more stable than SFT when using small batch sizes.
>
> # Concern 3:  Limited Analysis of Reward Function Design
> In our initial experiments, we actually explored a variety of reward functions and combination strategies beyond those reported in the main paper. Specifically, we performed a set of RL experiments on Qwen2.5-VL-3B-Instruct using the ChartQA dataset for training and evaluation.
>
> To assess the importance of the CERM reward, we conducted an ablation study where it was removed, leaving only the format reward. In this setting, model performance dropped to 0.48%, indicating that the model learned to follow the required response format but was not incentivized to produce correct answers.
>
> Beyond the format and CERM rewards, we experimented with an **Intermediate** reward designed to encourage the model to extract the accurate underlying data from chart images during the model’s step-by-step reasoning. Specifically, for every generated response, we parsed all numeric values within the <thinking...thinking> tags. We then computed two sub-scores:
>
> **Completeness**: The proportion of required data values the model was able to extract, calculated as 1 - |num_extracted - n| / n, where n is the number of expected values  (set to the number of numerical values in the chart’s underlying data table).
>
> **Correctness**: The percentage of these extracted values that exactly match entries in the ground-truth data table.
>
> The **Intermediate** reward was defined as the product of **completeness** and **correctness** scores. The intuition was to reward the model for both extracting all necessary values from the chart image and ensuring their accuracy before proceeding to the final answer.
>
> We incorporated this **Intermediate** reward in two ways: by simple addition with the other rewards (i.e., R_Format + R_CERM + R_Intermediate), and via custom weighting (e.g., 0.25 × R_Format + 0.5 × R_CERM + 0.25 × R_Intermediate). However, in practice, this led the model to focus too heavily on extracting correct data values, often at the expense of learning the mathematical reasoning skills required for complex questions (e.g., sum, average, difference) which are common in the benchmarks.
>
> | Reward Configuration                                      | ChartQA Performance (%) |
> | --------------------------------------------------------- | ----------------------- |
> | Format                                     | 0.48                    |
> | Format + CERM                                             | 85.8                    |
> | Format + CERM + Intermediate                              | 77.0                    |
> | 0.25 × Format + 0.5 × CERM + 0.25 × Intermediate | 80.88                   |
>
> These results demonstrate that the simple summation of the CERM and format rewards provides a very effective RL approach for our models. This is also consistent with the findings of concurrent works in the visual instruction tuning domain [3]. We will clarify these additional analyses and findings in the revised manuscript.

---

> > ### Author Response · Authors · 2025-06-03
> >
> > # Concern 4: Insufficient Dataset Construction Details
> > We provide further details on our dataset construction process below.
> >
> > **Dataset Sourcing and Balance**
> >
> > The composition shown in Figure 3 aims to reflect the natural distribution of chart types and visual styles found in real-world scenarios. For example, Google Search charts mainly include less common types (e.g., heatmaps, donut charts), which is why they make up only 8.6% of our dataset. Mint-1T (Common Crawl) and ArxivQA together represent 37.7%, as these sources are rich in scientific and academic charts. To ensure coverage of news reports and articles, we sampled 53.7% from ChartGemma, which is mainly sourced from news articles websites [2].
> >
> > **Filtering and Quality control**
> >
> > As described in Section 3.1.1 and detailed in Appendix A.2, we employed a two-stage filtering process using a ResNet-50 classifier for the Common Crawl (Mint-1T). The first stage used diverse, publicly available datasets (e.g., ChartQA, Chart-to-Text, AI2D, LRV-Chart for positives; CC-12M and ImageNet for negatives) to train the classifier without manual annotation. This model achieved 88% precision on a validation set. To further improve precision, we manually labeled 5,000 challenging samples and finetuned the classifier further on them. This increases classifier precision to 97%. In addition, all Google Search images (13K) underwent manual filtering after crawling to ensure high-quality chart selection. For established datasets such as ChartGemma, we used the data as provided since it had already undergone extensive filtering and quality control by its original authors.
> >
> > We will make sure to clarify these quality control, filtering, and normalization steps more explicitly in the main text of the revised manuscript to enhance reproducibility and transparency.
> >
> > # Concern 5: Presentation and Structural Issues
> > Thank you for pointing out these minor typos in the writing, we will fix them in the final revision of the manuscript.
> >
> >
> > Thank you for your thoughtful review. We hope our response has addressed your concerns and **kindly ask you to consider raising your score** to support the dissemination of our work within the ML community. Please let us know if you have any further questions.
> >
> > [1] DeepSeekMath: https://arxiv.org/pdf/2402.03300
> >
> > [2] ChartGemma: https://arxiv.org/abs/2407.04172
> >
> > [3] Visual-RFT: https://arxiv.org/pdf/2503.01785

---

> > > ### Comment · Reviewer_Lpvk · 2025-06-05
> > >
> > > I have read your responses. Thank you for providing additional information, which has addressed some of my concerns. Consequently, I will modify my score. It is important to clarify that this adjustment is based solely on the merit of your responses, rather than your request to "kindly consider raising your score".

---

> > > > ### Author Response · Authors · 2025-06-05
> > > >
> > > > Thank you for your response and for reconsidering your score. Could you please let us know your remaining concerns? We would be happy to provide further clarification or additional experiments if needed.

---

> > > > ### Author Response · Authors · 2025-06-10
> > > >
> > > > Dear Reviewer,
> > > >
> > > > Thank you for dedicating your time and effort to reviewing our manuscript and for providing insightful suggestions. As the author-reviewer discussion phase nears its conclusion, could you please let us know if you have any remaining concerns? We’re happy to provide additional clarification if needed.
> > > >
> > > > Best Regards,
> > > >
> > > > Authors

---

### Official Review · Reviewer_wCKY · 2025-05-16

**Rating:** 7
**Confidence:** 4
**Ethics Flag:** 1

**Summary:**

This work propose BigCharts-R1, where the author introduces two novel contributions: (1) A dataset curation pipeline to collect a large scale diverse chart plots from real-world resource and obtain accurate data-chart pairs via replot. (2) introduce a post-training pipeline SFT plus RL to augment general-purpose VLMs (Qwen-3B Vision and Qwen-7B Vision). Here the author introduces two verifiable novel reward functions designed toward chart understanding and reasoning tasks. Finally, the author has compared Big-R1 with a large set of Open-Source and Close-Source VLMs, including chart-based VLMs as well on ChartQA tasks. Big-R1 has achieved competitive performance across multiple benchmarks and further abalation study demonstrate that the proposed Reward Function plus GRPO leads to consistent performance gain on out-of-distribution dataset, which proves Reinformance Learning's capability to enhance model's generalization capability.

**Reasons To Accept:**

1.The author propose a very useful dataset resouce to augment VLM's capability for chart question answering tasks, which includes a combination of both synthetic charts and real-world resourced charts to assure a good divsersity. Meanwhile, it also propose a useful replot technique to address the inaccurate data extraction issue from real-world plots.
2. To my knowledge, this is the first post-training pipeline to include reinforcement learning for chart understanding tasks. The reinforcement learning stage significantly improve model's capability on numerical reasoning type chart understanding questions and the generalization to out-of-distribution tasks.
3. The details of data collection and model training are clearly introduced, which will help the community to follow up this line of research.

**Reasons To Reject:**

1. The experiment does not include a comprehensive study on the effectiveness of different components: (1) What is the usefulness of synthetic charts and real-world charts? (2) What is the effectiveness of the reploted charts vs simply using the original charts and the noisy underlined data prediction? (3) What is the effectiveness of the two reward function? How much gain or loss will I get if I remove one of them.
2. When compare to previous chart-specific LVLMs (ChartGemma-3B, TinyChart-3B), it would be more fair if those model are further fine-tuned on the BIgCharts datasets instead of simply comparing the original pre-trained weight.

---

> ### Author Response · Authors · 2025-06-03
>
> Thank you for your constructive comments. We address your concerns below:
> # Concern 1: Effectiveness of Different Components
> **Synthetic (Replotted) vs. Real-World (Original) Charts**
>
> As discussed in the manuscript (lines 37-39 and 45-50), real-world charts offer superior visual and topical diversity compared to synthetic charts, which typically lack such variety. However, real-world charts usually lack underlying metadata necessary for accurate Q/A generation, which is readily available for synthetic charts. Our proposed BigCharts approach addresses this issue by generating synthetic charts grounded in real-world data, thus preserving visual diversity while providing essential metadata for accurate Q/A pair creation.
>
> To validate our hypothesis, we conducted experiments comparing the performance of Qwen2.5-VL-3B trained using Q/A generated from original (real-world) charts directly versus our synthetic replotting approach that provide both the chart image and the underlying data. For consistency, Q/A pairs for the original charts were generated using Gemini Flash 2.0 and prompts similar to those used for the replotted charts (see Figure 6 in the Appendix). Both models were trained with identical hyperparameters as described in Section 5.1 to ensure a fair comparison. Results in the table below demonstrate that our replotted charts approach leads to improved performance, primarily due to more accurate Q/A generation enabled by metadata availability.
>
> | Method           | FigureQA Val 1 | FigureQA Val 2 | DVQA Easy | DVQA Hard | PlotQA T1 | PlotQA T2 | ChartQA Avg. | CharXiv Reas. | CharXiv Desc. | Avg.  |
> |------------------|----------------|----------------|-----------|-----------|----|---|---|---|---|-------|
> | Original Charts  | 73.70  | 73.10 | 75.20 | 72.50     | 72.40     | 52.40     | 82.24        | 30.90    | **63.62**  | 66.22 |
> | Replotted Charts | **76.10**  | **75.70**   | **76.30**     | **73.80**     | **74.60**     | **58.40**     | **84.60**      | **36.00**    | 62.85            | **68.71** |
>
>
> **Effectiveness of the Rewards Function**
>
> Our reward functions, the Format Reward and Chart Error Rate Reward (CERM), provide complementary benefits. The format reward promotes structured, step-by-step reasoning, while the CERM reward encourages the model to produce accurate numerical predictions. To assess their individual contributions, we conducted an ablation study by training the Qwen2.5-VL-Instruct model with the GRPO algorithm on the ChartQA dataset using different reward combinations.
>
> As shown in the table below, the CERM reward has the greatest impact on performance; removing it causes the model’s accuracy to collapse. The format reward mainly incentivizes the model to follow the required reasoning format, supporting step-by-step reasoning and better answer parsing. However, relying solely on the format reward is insufficient, as it does not incentivize correctness in the final answers, resulting in a significant drop in performance.
>
>
> | Rewards | ChartQA (Relaxed Accuracy) |
> | ---| ---|
> | Format | 0.48%   |
> | Format + CERM | 85.8% |
>
> # Concern 2: Fairer Comparison with existing works (ChartGemma, TinyChart)
>
> We appreciate this suggestion and would like to clarify that our original results (Table 1) demonstrate the advantage of fine-tuning on our proposed dataset (BigCharts) compared to previous datasets (TinyChart and ChartGemma). However, we agree that a more rigorous and controlled comparison, eliminating the bias introduced by different backbone models (Qwen2.5-VL, PaliGemma, TinyLLaVa), provides a fairer evaluation.
>
> To address this concern, we fine-tuned the same backbone model (Qwen2.5-VL-3B) on the publicly available ChartGemma and TinyChart datasets for one epoch and using the same hyperparameters reported in Section 5.1. For evaluation, we utilized the codebase and "program-of-thought" answer formats recommended by the authors of TinyChart and ChartGemma to ensure optimal and fair performance. The results in the table below confirm that fine-tuning with our BigCharts dataset consistently outperforms these datasets, particularly on challenging benchmarks such as ChartQA, PlotQA, and CharXiv.
>
> | Method                        | FigureQA Val 1 | FigureQA Val 2 | DVQA Val Easy | DVQA Val Hard | PlotQA T1 | PlotQA T2 | ChartQA Avg. | CharXiv Reas. | ChartarXiv Desc. | Avg.   |
> |-------------------------------|----------------|----------------|---------------|---------------|-----------|-----------|--------------|---------------|------------------|--------|
> | Qwen2.5-VL-3B (ChartGemma)    | 78.30   | 79.10  | 73.30         | 74.30         | 40.50     | 58.60     | 68.48   | 10.70   | 39.25  | 58.05  |
> | Qwen2.5-VL-3B (TinyChart)     | 47.30 | 48.80  | 51.40 | 48.19 | 57.09  | 50.00  | 70.28  | 24.60     | 41.62  | 48.80  |
> | **Qwen2.5-VL-3B (BigCharts)**     | 76.10  | 75.70   | 76.30   | 73.80 | 74.60  | 58.40     | 84.60        | 36.00   | 62.85  | **68.70**  |

---

> ### Comment · Reviewer_wCKY · 2025-06-09
> **Thanks for the detailed rebuttal**
>
> Many thanks for the author's details response to my original questions. My concern is properly addressed and I will still maintain my original score to this paper which I think is a great contribtution to the community.

---

### Decision · Program_Chairs · 2025-07-08

**Decision:**

Accept

**Comment:**

This paper investigates chart reasoning in VLMs and propose BigCharts, a dataset pipeline that re-generates real-world chart images with accurate underlying data, and BigCharts-R1, a model trained using supervised finetuning and GRPO-based reinforcement learning with chart-specific rewards. Experiments on five benchmarks demonstrate that BigCharts-R1 achieves state-of-the-art performance and significantly improves out-of-distribution generalization.

The paper received scores of 7, 4, 5, 6 before the rebuttal. After the authors’ response, both reviewers whom scored 4 and 5 raised their score, while the other reviewers maintained their original scores. There was also sufficient discussion between reviewers and authors, and the AC gave additional time for reviewers to further discuss.

Among reviewers and this AC, we acknowledge that this work proposes a solid "dataset curation strategy that combines real-world chart visual diversity with accurate underlying data through a replotting process", and that the supervised fine-tuning with GRPO-based RL is appropriate for addressing generalization issues in chart reasoning. The methodology is clearly described, along with a comprehensive empirical study, shows that the framework has "potential applicability to related tasks in visual-language understanding."

Some minor concerns remain, including:
- Evaluation centers on VQA; however, this AC considers VQA to be a complete and complex task itself.
- Some questions regarding the reward design for text answers; which can be further explored in future works.
- GRPO vs. PPO: authors are encouraged to include PPO experiments, but the presented GRPO-based approach is effective and well-motivated.

Based on the positive feedback after rebuttal and discussions, this AC recommends to accept this work and encourage the authors to incorporate the response and additional results to the main paper or supplementary materials.